# Structure analysis suggests Ess1 isomerizes the carboxy-terminal domain of RNA polymerase II via a bivalent anchoring mechanism

Kevin E. W. Namitz [1,3,5], Tongyin Zheng[2,5], Ashley J. Canning[1], Nilda L. Alicea-Velazquez[1,4], Carlos A. Castañeda [2✉], Michael S. Cosgrove[1✉] & Steven D. Hanes [1✉]

Accurate gene transcription in eukaryotes depends on isomerization of serine-proline bonds within the carboxy-terminal domain (CTD) of RNA polymerase II. Isomerization is part of the "CTD code" that regulates recruitment of proteins required for transcription and co-transcriptional RNA processing. *Saccharomyces cerevisiae* Ess1 and its human ortholog, Pin1, are prolyl isomerases that engage the long heptad repeat $(YSPTSPS)_{26}$ of the CTD by an unknown mechanism. Here, we used an integrative structural approach to decipher Ess1 interactions with the CTD. Ess1 has a rigid linker between its WW and catalytic domains that enforces a distance constraint for bivalent interaction with the ends of long CTD substrates (≥4–5 heptad repeats). Our binding results suggest that the Ess1 WW domain anchors the proximal end of the CTD substrate during isomerization, and that linker divergence may underlie evolution of substrate specificity.

[1] Department of Biochemistry and Molecular Biology, SUNY-Upstate Medical University, Syracuse, NY, USA. [2] Departments of Biology and Chemistry, Syracuse University, Syracuse, NY, USA. [3] Present address: Department of Chemistry, Pennsylvania State University, University Park, PA, USA. [4] Present address: Department of Chemistry and Biochemistry, Central Connecticut State University, New Britain, CT, USA. [5] These authors contributed equally: Kevin E. W. Namitz, Tongyin Zheng. ✉email: cacastan@syr.edu; cosgrovm@upstate.edu; haness@upstate.edu

Saccharoymces cerevisiae Ess1 (Essential in yeast 1) is the founding member of the eukaryotic parvulin-class of peptidyl prolyl cis–trans isomerase (prolyl isomerase; PPIase). Ess1 is highly conserved among eukaryotes[1–3] and plays a key role in transcription by regulating the activity of RNA polymerase II (RNAPII)[1]. However, the mechanism(s) by which Ess1 binds to RNAPII to carry out its function is not well understood. Specifically, it is not known how Ess1 engages the long unstructured carboxy-terminal domain (CTD) of Rpb1, the largest subunit of RNAPII.

Ess1 and other prolyl isomerases (cyclophilins, FK506-binding proteins) regulate the folding and activity of target proteins by catalyzing a 180° rotation of the peptide bond preceding proline, causing conformational changes[4–6]. Ess1 isomerizes the CTD of Rpb1, facilitating the recruitment of proteins needed for efficient transcription and RNA processing[7–9]. Loss of Ess1 has widespread deleterious consequences on RNAPII transcription, including cryptic transcription and defects in elongation, termination/3'-RNA-processing and histone modification[9–13]. Pin1, the human ortholog of Ess1[2], is also implicated in regulation of RNAPII transcription[14]. Both Pin1 and the Drosophila melanogaster ortholog of Ess1 (called Dodo), can substitute for Ess1 in yeast, indicating functional conservation[2,3].

The Rpb1 CTD is composed of an unstructured heptad repeat with a consensus sequence of $Y_1-S_2-P_3-T_4-S_5-P_6-S_7$. There are 26 repeats in yeast (nearly all consensus), and 52 repeats in humans (about half consensus)[15,16]. In humans, the divergence is most pronounced in the second half of the CTD, where substitutions at position 7 are most frequent (S > K). Despite this divergence the two S-P motifs are nearly invariant. Phosphorylation of Ser2 or Ser5 within the CTD repeat generates Ess1/Pin1-binding sites (pSer-Pro motifs)[17,18]. In vitro, Ess1 increases cis/trans isomerization of the pSer2–Pro3 and pSer5–Pro6 bonds from a spontaneous rate of <1 turnover/min to about 200 and 1000 turnovers/min, respectively[18]. Importantly, Ess1 and other prolyl isomerases act reversibly, providing a kinetic effect, which can effectively increase the availability of the less abundant (~10%) cis-isomers[19–21]. We have proposed a "traffic cop model" whereby Ess1 maintains the CTD in a conformationally dynamic state to promote timely co-factor exchange, thus increasing the efficiency of multiple steps of the transcription cycle[1].

Ess1 and Pin1 are small modular proteins (~19.5 kDa) composed of two compact domains; an N-terminal Type-IV WW domain[22] and C-terminal catalytic (PPIase) domain. Both domains bind pSer/pThr-Pro motifs, with the WW domain having ~10-fold higher affinity[23]. How these domains work together is not known. This is a particularly vexing question because physiologically relevant substrates, such as the CTD, usually contain multiple pSer-Pro-binding motifs. A number of mechanisms including sequential binding, competitive binding, and cooperative binding have been proposed[23–26].

Structure studies of Pin1, and Ess1 from the fungal pathogen Candida albicans (CaEss1), show the WW and PPIase domains are similar, but that the linker region that joins them are different[27,28] (for sequence alignments, see Fig. S1). In Pin1, the linker is unstructured and there are dynamic interactions between the WW and PPIase domains[29,30], which may contribute to Pin1's broad substrate specificity[31–33]. By contrast, the highly structured linker in CaEss1 restricts the relative orientation between the WW and PPIase domains[34]. These differences might impact the way in which Pin1 and CaEss1 engage multivalent substrates. To date, there is no structural information on the S. cerevisiae Ess1, and it is not known if highly structured linker regions are a common feature of the fungal Ess1 enzymes.

To better understand how Ess1 recognizes long, physiologically relevant substrates like the CTD, we determined the crystal structure of S. cerevisiae Ess1 and studied its interaction with a series of bivalent CTD peptides of increasing length. The WW and catalytic domains of Ess1 are similar to that of human Pin1 and CaEss1, however the linker region and the relative orientation of the two domains is different. Together with solution studies, our results indicate that Ess1 has an elongated structure with a highly structured linker with a short α-helix. Binding studies using analytical ultracentrifugation, fluorescence anisotropy, and NMR chemical shift analyses revealed simultaneous and potentially cooperative interaction of the Ess1 WW and catalytic domains with long bivalent substrates. The results are the first to identify bivalent Ess1–CTD interactions, suggesting an anchored mechanism of isomerization, and raising the possibility that during evolution, eukaryotic parvulin-class PPIases gained a broader substrate specificity by acquiring a flexible linker that generates a more dynamic (and promiscuous) binding mode.

## Results

**Overall structure of budding yeast Ess1.** S. cerevisiae Ess1 (henceforth called Ess1) was co-crystallized with a single heptad repeat (1R) phospho-Ser5 CTD peptide and the X-ray structure was determined at 2.4 Å resolution (Fig. 1a, Table 1). The globular domains of Ess1 are similar to those in human Pin1[28] and CaEss1[27] (Fig. 1a). The Ess1 N-terminal WW domain (residues 10–45) forms a three stranded anti-parallel β-sheet as described for Type IV WW-domains that recognize phospho-Ser-Pro motifs[22], and is highly similar to those in Pin1 and CaEss1, superimposing with an RMSD of 0.7 Å (Fig. 1b). A peak of positive electron density in the WW domain near W38 and Y27 (Fig. S2a) was observed that we interpret as the position of proline 6 of the heptad repeat, as observed in the Pin1-CTD structure[23]. The remainder of the CTD peptide was disordered and could not be modeled. In addition, there was no evidence of CTD peptide binding to the PPIase domain, which is likely due to its lower affinity for the PPIase domain compared to that of the WW domain[23].

The PPIase domain of Ess1 (aa 64–170) has a globular α/β-fold structure nearly identical to that of Pin1 and CaEss1, which superimposes with RMSDs of <0.7 and 0.6 Å, respectively (Fig. 1c). The PPIase domain consists of an anti-parallel β-sheet that forms a concave surface bordered by α3 and α5 helices of the PPIase domain, forming the catalytic core. The entrance to the active site in Ess1 is formed by a large loop (aa 68–85) that includes basic residues K68, R73, R74 (Fig. S2b), similar to those in Pin1 (K63, R68, R69) that interact with the phosphate group of pSer-Pro substrates[28,35]. In the Ess1 structure, the basic loop is in the "closed" conformation as in CaEss1[27] and the original Pin1 structure[28], rather than in an "open" conformation seen in a later Pin1 structure[23]. Key catalytic site residues identified in Pin1 (including H59, C113, H157, S115, Q131, F134)[28,35–38] are conserved in Ess1 (H64, C120, H164, S122, Q138, F141) and located in analogous positions (Figs. 1d, S1, S2c). In summary, the WW and PPIase domains in Ess1, CaEss1 and human Pin1 are individually nearly identical, consistent with their similar binding preferences (pSer-Pro), and the functional interchangeability of these proteins in yeast[2,3].

**Structural differences between ScEss1, Pin1, and CaEss1.** The overall shape of Ess1, CaEss1, and Pin1 show striking differences in the relative spatial positions of the WW and PPIase domains (Figs. 1a and S2b). With the catalytic domains aligned, the WW domain adopts a distinct position in Ess1, CaEss1, and Pin1. The Pin1 WW domain is close to the 5-turn α1-helix of the PPIase domain, forming a pocket where a CTD peptide binds[23]. In contrast, the WW domain in CaEss1 is positioned up and away

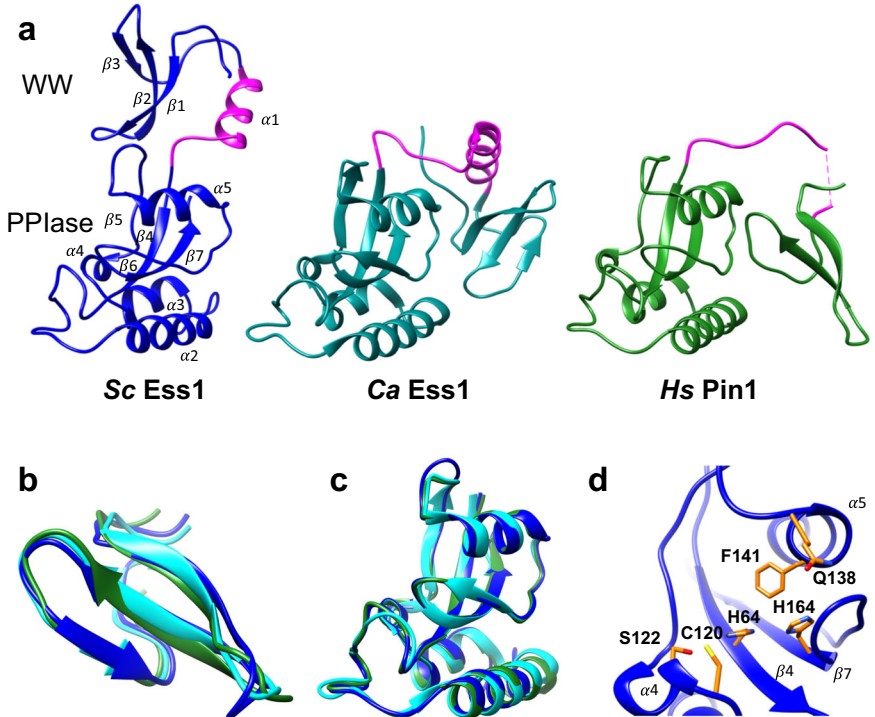

**Fig. 1 The crystal structure of the *S. cerevisiae* Ess1 protein reveals an elongated protein with a well-ordered linker joining the WW and catalytic domains. a** Comparison of *S. cerevisiae* Ess1, residues 9–170, (blue), *C. albicans* Ess1 (PDB ID: 1YW5) (cyan) and human Pin1 (PDB ID: 1PIN) (green) highlighting the different relative positions of the two functional domains. Linker regions are highlighted (pink). **b** Superposition of the WW domains (Cα) of each protein shown in (**a**). The overall degree of similarity is very high (RMSD < 0.7 Å). **c** Superposition of the PPIase (catalytic) domains (Cα) of each protein in (**a**) shows a high degree of similarity (RMSD < 0.7 Å). **d** Close-up of the catalytic site within the *Sc*Ess1 PPIase domain, with critical residues highlighted. Each of these residues in *Sc*Ess1 (H64, C120, H164, S122 Q138, F141) is conserved in *Hs*Pin1 (see text).

## Table 1 Data collection and refinement statistics[a].

| | *Sc*Ess1 (PDB ID: 7KKF) |
|---|---|
| *Data collection* | |
| Space group | C2 |
| Cell dimensions | |
| *a, b, c* (Å) | 110.1, 57.4, 69.3 |
| *α, β, γ* (°) | 90.0, 96.9, 90.0 |
| Resolution (Å) | 50.00–2.39 (2.54–2.39)[b] |
| $R_{sym}$ | 0.099 (0.712) |
| $I / \sigma I$ | 34.9 (3.4) |
| Completeness (%) | 99.8 (90.0) |
| Redundancy | 7.0 (7.0) |
| *Refinement* | |
| Resolution (Å) | 29.21–2.40 (2.55–2.40) |
| No. of reflections | 16,741 |
| $R_{work}/R_{free}$ | 0.269/0.306 |
| No. of atoms | |
| Protein | 2344 |
| Ligand/ion | – |
| Water | 28 |
| *B*-factors | |
| Protein | 56.8 |
| Ligand/ion | – |
| Water | 52.1 |
| R.m.s. deviations | |
| Bond lengths (Å) | 0.009 |
| Bond angles (°) | 1.357 |

[a]X-ray diffraction data were generated from a single crystal.
[b]Values in parentheses correspond to highest-resolution shell.

from this helix and engages in numerous interdomain contacts with the PPIase domain not seen in Pin1[34]. Finally, in the Ess1 structure, the WW is even further removed and occupies a space on the opposite side from the PPIase domain α1-helix, generating an elongated structure.

To determine if this domain orientation on Ess1 is preserved in solution, we measured small angle X-ray scattering (SAXS) profiles of dilute solutions of Ess1 (Fig. S3, Table S1). The refined average three-dimensional ab initio molecule envelope showed that Ess1 has an elongated (peanut-shaped) structure in which the crystal structure fits well (Fig. 2a). In addition, the scattering curve calculated from the crystal structure of Ess1 provides a better fit to experimental SAXS data ($\chi = 0.21$) compared to that of CaEss1 ($\chi = 0.96$) and Pin1 ($\chi = 1.47$) (Fig. 2b). These results indicate that the elongated shape of Ess1 observed in the crystal structure is also the predominant form in solution.

This position of the WW domain in Ess1 differs from that of CaEss1 by a 270° clockwise rotation about the loop C-terminal to the linker helix (Fig. S2b). In this position, only limited contacts between the WW and PPIase domains in Ess1 are observed: water-mediated hydrogen bonds between WW residue Y19 and PPIase residue E136, and WW residue K24 to the backbone carbonyl of PPIase W131. In addition, the carbonyl of WW K24 forms a hydrogen bond with R59 of the linker. Whether these contacts would be sufficient to immobilize the WW domain in Ess1 is not known.

The distinct arrangements of the WW and PPIase domains in the three Ess1 orthologs likely derive from their distinct linker regions. In Pin1, the linker (S38-R54) is disordered and the motions between the individual domains are not constrained[29,30,35]. Only in the presence of substrate do the Pin1 WW and PPIase domains orient to form a

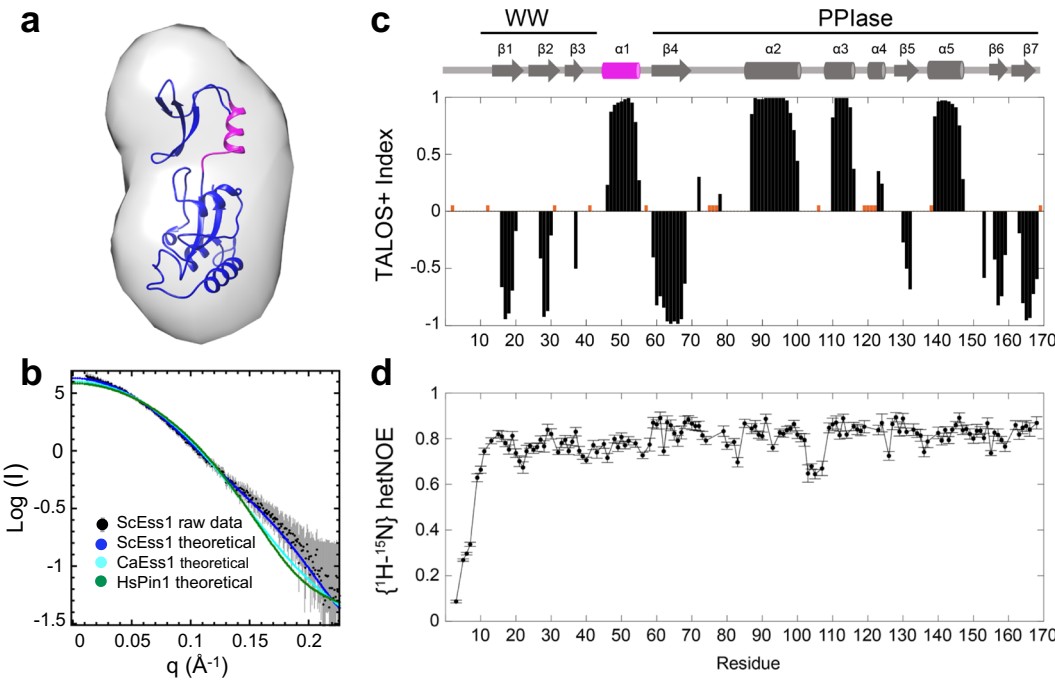

**Fig. 2 SAXS and NMR data show Ess1 is elongated, and fully ordered in solution, including the linker. a** The Ab initio small angle X-ray scattering (SAXS) envelope overlaid with the crystal structure. Envelope is contoured at 20 Å resolution. **b** Scattering profile of ScEss1 at 4.78 mg/mL (black points; gray error bars) was used for comparison of SAXS profiles predicted from the crystal structures of HsPin1 (green points), CaEss1 (cyan points), and ScEss1 (blue points) in the program Crysol (ATSAS package, EMBL). The Chi values for each of these fits to the raw data are as follows: 1.467 (HsPin1), 0.959 (CaEss1), and 0.209 (ScEss1), indicating the best fit for the ScEss1 structure. **c** TALOS+ secondary structure calculation using NMR backbone HN, N, CO, CA, and sidechain CB chemical shift assignments. TALOS confirms the α-helical structure of linker (residues 46–59 in pink). **d** Backbone $^1$H-$^{15}$N heteronuclear NOEs show that the protein is well-structured throughout the WW and PPIase domains, including the linker region. Errors in NOEs were determined using standard error propagation (see the "Methods" section).

binding pocket[29,30,39]. In contrast, the linker regions in Ess1 (K46-R59) and CaEss1 (K43-Q67) are highly ordered and contain a solvent-exposed, amphipathic α-helix[27,34] (Fig. 1a, colored in magenta). The Ess1 linker α-helix is shorter (~3 turns) than in CaEss1 (4 turns), however, its position relative to the WW domain is the same. There are no apparent contacts between residues in the Ess1 linker and the PPIase domain, unlike those observed in CaEss1[27].

**NMR studies confirm Ess1 has an elongated, conformationally constrained shape with a structured helical linker.** To obtain structural and dynamic information about Ess1 in solution on an atomic level, we used nuclear magnetic resonance (NMR) spectroscopy, where we obtained 92% of backbone $^1$HN, $^{15}$N, $^{13}$Cα, $^{13}$CO and sidechain $^{13}$Cβ chemical shift assignments for a C120S variant of Ess1. These assignments were then visually transferred to NMR spectra of wild-type Ess1 for all subsequent experiments discussed below (see the "Methods" section, Fig. S4). Using TALOS+ predictions[40], we confirmed the secondary structure elements of Ess1 (Fig. 2c). including the α-helical structure of the linker (Fig. 2c).

We analyzed Ess1 backbone dynamics using standard $^{15}$N $R_1$, $R_2$, and hetNOE experiments[41]. The relaxation rates and hetNOE data suggest that the majority of residues in Ess1 are in well-defined structural elements, including the linker region (K46-R59) (Figs. 2d and S5). Using the $^{15}$N backbone relaxation NMR data, we calculated the rotational diffusion tensors for the individual domains in Ess1 as well as for the entire protein (Table S2). The diffusion tensor characteristics for the individual domains are nearly identical to that of the full-length protein, suggesting that the WW and PPIase

domains tumble as a single, associated unit. Furthermore, the rotational correlation time ($\tau_c$) of 13.5 ns is consistent with the expected $\tau_c$ of a compact, monomeric 20 kDa globular protein[42]. These data corroborate the SAXS measurements and analytical ultracentrifugation data discussed below.

In summary, Ess1 is relatively rigid in solution, similar to CaEss1[34], but different from human Pin1, which is flexible and whose WW and PPIase domains tumble relatively unconstrained until substrate binds[30]. The key differences between fungal Ess1 proteins and mammalian Pin1 map to the distinct linker regions that join the highly conserved functional domains. The fungal linkers are highly structured and constrain the WW and PPIase domains, resulting in a more rigid structure than in the mammalian enzyme. Secondary structure predictions based on fungal and metazoan sequences are consistent with this idea[1,27]. This divergence between fungal and metazoan Ess1/Pin1 proteins may have implications for both binding mechanisms and substrate specificities (see also refs. [1,27]).

**How do Ess1 and Pin1 bind to multivalent CTD substrates?** Both the WW and PPIase domains of Ess1 and Pin1 bind pSer/pThr-Pro motifs. This dual-binding capacity, and the fact that most substrates contain multiple binding motifs complicates affinity measurements. Deciphering the mechanism of action of these proteins has been challenging and controversial. Early models suggested the WW domain tethers Ess1/Pin1 to protein substrates, increasing the local concentration of the PPIase domain, which then isomerizes nearby pSer-Pro motifs. This is based on the ≥10-fold higher binding affinity of the WW domain for single-site peptides in vitro[17,23]. However, it is not known

**Table 2 Peptides used in this study.**

| Name | Sequence | Source |
|---|---|---|
| BLI-1R | biotin-GGSGGS(YSPT**pSP**S)YS | NeoBioLab |
| FITC-1R | FITC-AS(YSPT**pSP**S)YS | Genscript |
| FITC-2R | FITC-AS(YSPT**pSP**S)(YSPT**pSP**S)YS | Abclonal |
| FITC-3R | FITC-AS(YSPT**pSP**S)(YSPTSPS)(YSPT**pSP**S)YS | Abclonal |
| FITC-4R | FITC-AS(YSPT**pSP**S)(YSPTSPS)(YSPTSPS)(YSPT**pSP**S)YS | Genscript |
| FITC-5R | FITC-AS(YSPT**pSP**S)(YSPTSPS)(YSPTSPS)(YSPTSPS)(YSPT**pSP**S)YS | Abclonal |
| NMR-1R | AS(YSPT**pSP**S)YS | Abclonal |
| NMR-2R | (YSPT**pSP**S)(YSPT**pSP**S) | Abclonal |
| NMR-3R | (YSPT**pSP**S)(YSPTSPS)(YSPT**pSP**S) | Abclonal |
| NMR-4R | AS(YSPT**pSP**S)(YSPTSPS)(YSPTSPS)(YSPT**pSP**S)YS | Abclonal |
| NMR-5R | AS(YSPT**pSP**S)(YSPTSPS)(YSPTSPS)(YSPTSPS)(YSPT**pSP**S)YS | Abclonal |

Heptad repeat units are indicated in parentheses, and phosphoserine-proline binding motifs are highlighted in bold. Phosphate modifications were incorporated during synthesis (see the "Methods" section for details). BLI-1R peptide was used for co-crystallization.

whether the WW and PPIase domains bind multisite (multivalent) substrates simultaneously and/or cooperatively, or whether the domains compete with each other for occupancy. Nor has the stoichiometry and arrangement of Ess1/Pin1 proteins on long, physiologically relevant substrates been determined. To address these questions and gain a mechanistic understanding of how Ess1 interacts with its major in vivo target, the Rpb1 CTD, we determined the affinity and stoichiometry of Ess1–CTD interaction using multiple orthogonal approaches.

**Ess1 binds better to longer CTD peptides.** Prior studies of Ess1/Pin1–CTD interaction were limited to peptides bearing only a single heptad repeat ($Y_1S_2P_3T_4S_5P_6S_7$). To provide a more realistic model of CTD interaction, we generated a series of CTD peptides of increasing length ranging from 1 to 5 heptad repeats (1R–5R) (Table 2). To simplify the analysis, phosphorylation (incorporated during synthesis) was restricted to only the outermost repeats, and positioned exclusively on the Ser5-Pro6 motif. The pSer5-Pro6 position was chosen because it has a higher binding affinity and turnover rate than does pSer2-Pro3[18,23], and because mutations of Ser5 show a stronger genetic interaction with Ess1 in vivo[12].

To estimate the affinity of different length CTD peptides, we used a competition fluorescence anisotropy assay in which we measured the ability of unlabeled 1R–5R CTD peptides to compete for Ess1 binding with an FITC-labeled 1R-CTD peptide (Fig. 3, Table 3a). As expected, a 2R peptide ($IC_{50} = 59 \pm 17 \mu M$) and 3R peptide ($IC_{50} = 60 \pm 17 \mu M$), which have two binding sites, competed better than the control 1R peptide ($IC_{50} = 259 \pm 33 \mu M$). However, the 4R ($IC_{50} = 35.9 \pm 0.5 \mu M$) and 5R ($IC_{50} = 41 \pm 5.0 \mu M$) peptides competed even better, despite all having two binding sites. The results are consistent with a model in which Ess1 occupies both sites simultaneously on the longer CTD-peptide substrates (4R, 5R).

**Ess1 binds as a monomer, favoring 5R-CTD peptides.** Ess1–CTD interactions were also analyzed using sedimentation velocity analytical ultracentrifugation (SV-AUC), a method that maintains the equilibrium between free and bound species as the complex sediments in a gravitational field[43]. As such, it is possible to measure binding affinities, stoichiometry, cooperativity and potential conformational changes upon binding[43]. Ess1 is a stable globular protein that sediments as a monodisperse monomer that did not change over a 4-fold concentration range (Fig. 4a). The sedimentation coefficient of Ess1 ($s \sim 1.45$; $f/f_0 \sim 1.35$) indicates there is added hydrodynamic drag consistent with an elongated shape in solution, vs. a spherical protein of this size, which would

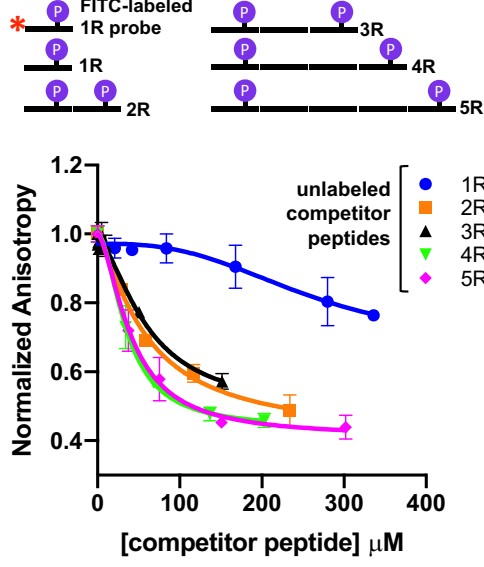

**Fig. 3 Fluorescence anisotropy measurements indicate that Ess1 binding favors longer CTD substrates.** A competition assay was performed using unlabeled 1R–5R CTD peptides to compete with an FITC-labeled 1R-CTD peptide (1 μM) for Ess1 (50 μM) binding. A 1 repeat CTD peptide labeled with FITC at its N-terminus was used as a probe (FITC-1R, Table 2). Competition was carried out using increasing concentrations of unlabeled 1R-5R CTD peptides phosphorylated at the Ser5 positions of the terminal heptad repeats (see "NMR-1R to NMR-5R" peptides, Table 2). Each point represents an average (±SD) of $n = 2$ or 3 from independent experiments. $IC_{50}$ values were calculated and shown in Table 3.

have a higher $s$ value ($s = 1.65$; SEDNTERP)[44], consistent with the SAXS and NMR results.

Unbound FITC-labeled CTD peptides did not sediment appreciably ($s \sim 0.0$–0.5), as monitored by absorbance at 490 nm (Fig. 4b). However, upon addition of Ess1, the majority of a 1R CTD peptide sedimented ($s \sim 1.5$) coincident with monomeric Ess1 (Fig. 4b). Sedimentation analysis for 4R with Ess1 also suggested the majority of the peptide is bound by Ess1 (shifted to $s > 1.5$) (Fig. 4c). Interestingly, the proportion of shifted 5R peptide (Fig. 4d) is higher than that of 4R peptide, indicating a more favorable, potentially cooperative interaction of Ess1 with the 5R peptide.

To better understand the mechanism of interaction, we titrated Ess1 into a fixed amount of each peptide and integrated each distribution to determine the signal-weighted average sedimentation

**Table 3 Summary of Ess1-CTD binding affinities.**

**a. Competition anisotropy (vs. FITC-1R)**

| Competitor peptide | IC$_{50}$ ($\mu$M) | SD (standard deviation) ($\mu$M) |
|---|---|---|
| 1R | 261 | 33 |
| 2R | 59 | 17 |
| 3R | 60 | 17 |
| 4R | 36 | 0.5 |
| 5R | 41 | 5 |

**b. Titration by SV-AUC**

| Ligand | Two-site model | |
|---|---|---|
| | K$_{d1}$ ($\mu$M) | K$_{d2}$ ($\mu$M) |
| FITC-1R | 279 | >14,000 |
| FITC-4R | 101 | 8330 |
| FITC-5R | 21.5 | 5990 |

**c. Titration by NMR**

| Ligand | Average $K_d$ ($\mu$M) | SD ($\mu$M) |
|---|---|---|
| 1R | 154.7 | 44 |
| 4R | 46.8 | 74 |
| 5R | 9.1 | 8 |

(a) Fluorescence anisotropy measurements of binding avidity of 1R, 2R, 3R, 4R, and 5R peptides using a competition assay. IC$_{50}$ values were calculated with Sigmaplot using a four parameter logistic curve model. Note that the concentration of Ess1 is 50-fold higher than the FITC-1R-CTD peptide being competed, which is likely all bound to the WW domain. (b) Binding affinities of Ess1 for 1R, 4R and 5R FITC-labeled peptides as determined by SV-AUC $s_w$ isotherm analysis. (c) Overall binding affinities of Ess1 for 1R, 4R, and 5R peptides determined using NMR titrations (see the "Methods" section).

coefficient ($s_w$), which was plotted against Ess1 concentration (Fig. 4e). To fit the data, we constrained the minimum and maximum $s_w$ values to the same range for each peptide, and simulations were performed using Kintek Explorer software[45] using a two-site binding model. Good fits were obtained for the 5R and 1R CTD peptides, which showed that 5R bound with an affinity for the first site that was an order of magnitude greater than that of the 1R peptide (Table 3b). Second site binding affinities also showed a preference for 5R (and 4R) over 1R, but with much weaker affinities and with smaller differences among the peptides (Table 3b). Fitting of the 4R-CTD peptide with the same model produced a poor fit to the data (Fig. 4e, black dashed line). However, releasing the constraint for the maximum $s_w$ value significantly improved the fit for the 4R titration, and resulted in an intermediate $K_d$ between that of 1R and 5R (Fig. 4e, black line; Table 3). This indicates that the hydrodynamic properties of the 4R:Ess1 complex are distinct from that of the 1R- and 5R-complexes, and likely reflects conformational differences in Ess1 upon binding the 4R peptide. Thus, binding of Ess1 to the 4R peptide, while better than that of 1R, is somewhat compromised in comparison to binding to the 5R peptide. These and NMR data presented below are consistent with the idea that binding interactions with the 5R peptide are more favorable than with the 4R peptide, in which some protein distortion may occur to enable simultaneous binding by the WW and PPIase domains (Fig. 4f).

Importantly, for both 4R and 5R peptides, there is no indication of a 2:1 protein:peptide stoichiometry, as might be expected if two Ess1 molecules bound to the same peptide, one at each end. If this occurred, the resulting ternary complex would likely increase in mass by ~19.5 kDa (the size of an Ess1 monomer) and appear as a peak with higher sedimentation value (s ~3). Therefore, the simplest interpretation is that the 4R and 5R complexes have a 1:1 (Ess1:CTD peptide) stoichiometry. We cannot formally rule out a 1:2 ratio, where a different peptide binds to each of the two protein domains. For the smaller peptides, particularly the 1R and 2R, this scenario might be difficult to detect due to their relatively small molecular weights for SV-AUC. However, we do not favor this interpretation because the experiments were done with limiting peptide concentrations (10:1 molar excess of protein).

**WW and PPIase domain contacts are enhanced with longer CTD peptides.** To identify individual residues associated with the binding interface on Ess1, we titrated unlabeled CTD peptides and used NMR to monitor the backbone amide chemical shifts of residues in Ess1. From these spectra, we calculated chemical shift perturbations (CSPs) and mapped these onto the structure of Ess1 (Fig. 5a). For the 1R peptide, residues with the strongest CSPs mapped almost exclusively to the WW domain, including residues S20, K21, S22, K23, Y27, F29, S36 and E39 (Fig. 5a). These residues correspond to the same residues in the Pin1 WW domain (S16, R17, Y23, F25, S32) that interact with a single CTD repeat[23,46,47]. Unlike in Pin1, however, residues 130–137 in the PPIase domain of Ess1 were also perturbed in the presence of the 1R peptide. These residues are spatially close to the WW domain, thus we suspect that the observed CSPs are sensitive to CTD binding in the WW domain (Fig. 5a). CSPs for the 2R and 3R peptides (taking into account the differences in stoichiometry of binding sites) were generally similar to 1R results, both in terms of CSP magnitudes and overall CSP pattern (Fig. S6).

Interestingly, addition of the 4R and 5R peptides resulted in larger CSPs in the WW domain, suggesting stronger overall binding (Fig. 5a). These included residues 25–29 and 34–41. Higher CSPs were also observed with 4R and 5R peptides in a PPIase patch consisting of residues D102-S118 and R125. These residues are not in the active site of the catalytic domain, but could be involved in stabilizing binding of the two-site simultaneously bound peptides (see below). CSPs in the active-site region were relatively minor, but include residues E136, E142, S159 and G162, which overlay with residues in the Pin1 active site engaged with peptide mimetics (PDB ID 3TCZ)[19]. These experiments suggest that the longer CTD peptides (4R, 5R) enhance contacts with both WW and PPIase domains, simultaneously.

**Five CTD repeats is the minimal optimal length for Ess1 binding.** To determine Ess1 binding $K_d$s on a residue-by-residue basis, we performed NMR titration experiments using 1R, 4R, and 5R CTD peptides (Table 3c, overall $K_d$s; Table S3, $K_d$s for all residues, Fig. S7). The results suggest that the 1R CTD binds preferentially to the WW domain of Ess1, as most of the CSPs localize to this region. Using a single-site binding model on residues with CSPs > 0.03 ppm at the titration endpoint, we determined that the $K_d$ was 154.7 ± 44 $\mu$M, in general agreement with FA and SV-AUC results above (and a published $K_d$ of ~60 $\mu$M)[18]. While we observed some CSPs near or at the active site of the PPIase domain, they were generally < 0.05 ppm for these residues (e.g. K70, T84, S159, G162). As others have reported weak peptide binding to the PPIase domain (>500 $\mu$M)[23], our NMR titration experiments, which used 100 $\mu$M Ess1 protein, would not be sensitive to such weak binding. Therefore, the observed CSPs in the PPIase domain are either weakly reporting on structural perturbations resulting from 1R binding to the WW domain, or on transient interactions between the 1R and the active site.

In the presence of 4R or 5R peptides, we observed larger CSPs in the WW domain as well as near the PPIase active site, specifically residues E111 and R125 (residues E104 and A118 in Pin1). In addition, a number of WW residues underwent intermediate exchange during the titration, indicative of stronger binding to 4R or 5R (Fig. S8). Indeed, the NMR titration with the 5R peptide indicated that residues throughout the protein (both WW and PPIase domains) titrated with significantly stronger binding affinity of 9.1 ± 8 $\mu$M compared to 1R when assuming a single-site binding model. These binding affinities are consistent with those observed by SV-AUC (Table 3c). The results for the 4R titrations were intermediate to those of the 1R and 5R (Fig. 5b),

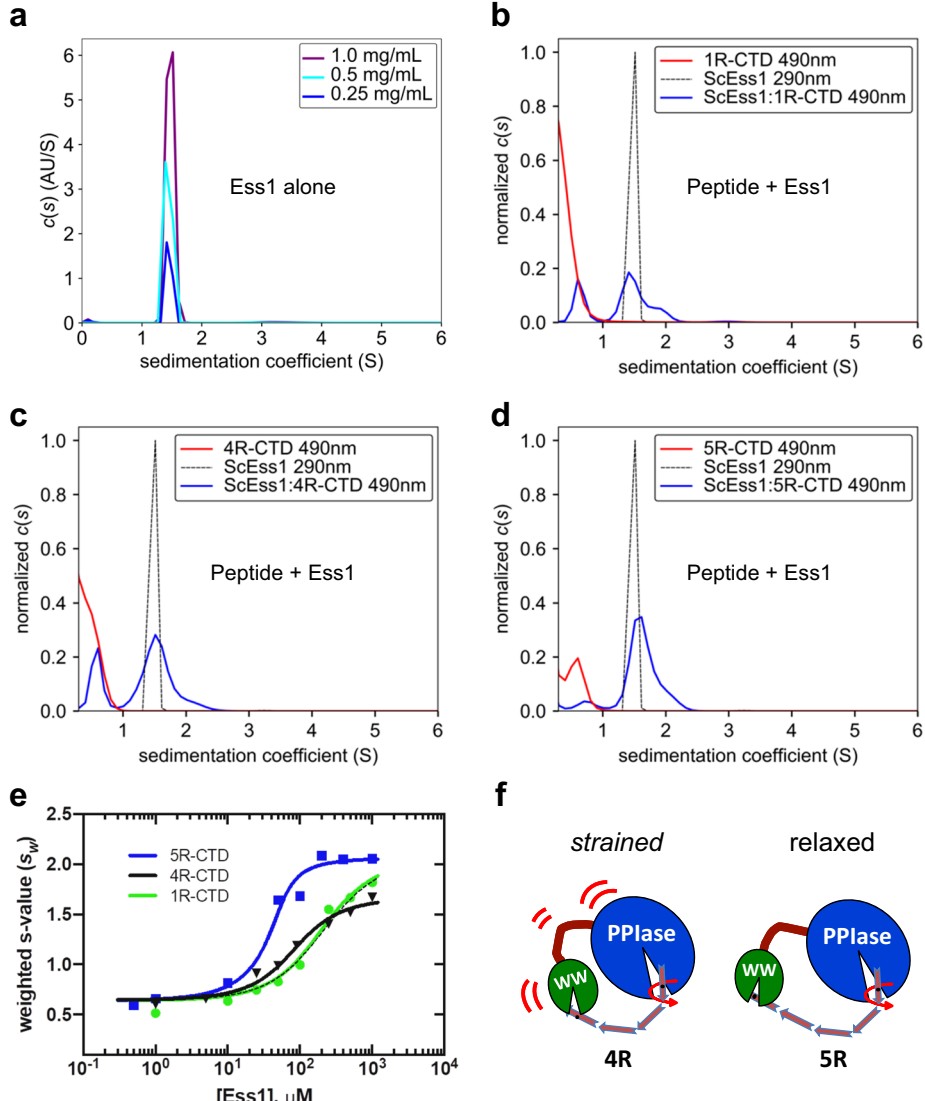

**Fig. 4 SV-AUC reveals potential cooperative binding of Ess1 to a 5-repeat CTD peptide. a** Ess1 alone at three different concentrations, showing it is a stable, monodisperse monomer. **b–d** Binding of Ess1 to FITC-labeled CTD peptides as revealed by $c(s)$ distributions: **b** FITC-**1R**-CTD peptide, **c** FITC-**4R**-CTD peptide, and **d** FITC-**5R**-CTD peptide. Each panel shows the position of 200 μM Ess1 alone at 290 nm (black dotted line), or 20 μM CTD peptides alone (red line) or a mixture of Ess1 + CTD peptide at 10:1 (blue line) at the indicated wavelenths of absorbance. Note the larger fraction of the 5R peptide migrating at the bound position vs. the 4R peptide. **c** vs. **d e** $s_w$-isotherms of 50 μM 1R-CTD (green), 4R-CTD (black) or 5R-CTD (blue) with a titration of ScEss1 (0.5–1000 μM). The black dashed line shows the fit to the 4R-CTD data with the constrained maximum $s_w$ value (see text). **f** Cartoon model of Ess1 binding to 4-repeat and 5-repeat CTD peptides phosphorylated at the terminal Ser5-Pro6 repeats. The model is based on the higher affinity of the 5R peptide compared to the 4R peptide (panels **c** vs. **d**; **e**), and supported by SV-AUC and NMR chemical-shift data (see text).

with an average $K_d$ of 46.8 ± 74 μM. Curiously, variable 4R-binding affinities were reported across the protein, giving rise to the high standard deviation for the $K_d$ value (Fig. 5b). While most of the residues in the WW domain titrated with 5R-like binding affinity, residues K23 and H35 did not, together with PPIase residues D103, N108, D119, and Y123 (Fig. 5b, Table S3). Importantly for the 4R titration, we observed non-linear or distinct (from 1R and 5R) chemical shift trajectories for WW residues N30 and H35, linker α-helix residue L49, and PPIase residues G130 and D143 (Figs. 5c and S9), suggesting a secondary event occurs during the course of the titration. We suspect that the 4R peptide is slightly shorter than the optimum length for two-site binding, and consequently Ess1 may undergo a slight conformational change across the protein to accommodate it (Fig. 4f), in agreement with SV-AUC data presented above. This would also explain the lower apparent binding affinity for 4R vs.

5R in the SV-AUC analysis (Fig. 4e). Notably, all chemical shift trajectories were linear in the presence of 5R, consistent with the idea that no substantial conformational change occurs in Ess1 when interacting with 5R-CTD.

**A model of anchored prolyl isomerization**. To better understand the interactions between Ess1 and the CTD, we modeled binding of different length CTD peptides. We docked a 1R-CTD peptide in the Ess1 WW domain (Fig. 6a) based on a prior co-crystal structure of the Pin1-WW with a CTD peptide (PDB 1F8A)[23]. Next, we docked a 1R-CTD peptide in the Ess1 PPIase active site (Fig. 6a) based on the pSer-Pro motif of a peptidomimetic inhibitor in Pin1 (PDB 3TCZ)[38]. The docking shows that a 2 or 3 repeat CTD could not reach both WW and PPIase-binding sites, consistent with our affinity measurements from FA, SV-AUC and NMR. Modeling 2R-CTD peptides into each of the WW and PPIase domains indicates that a

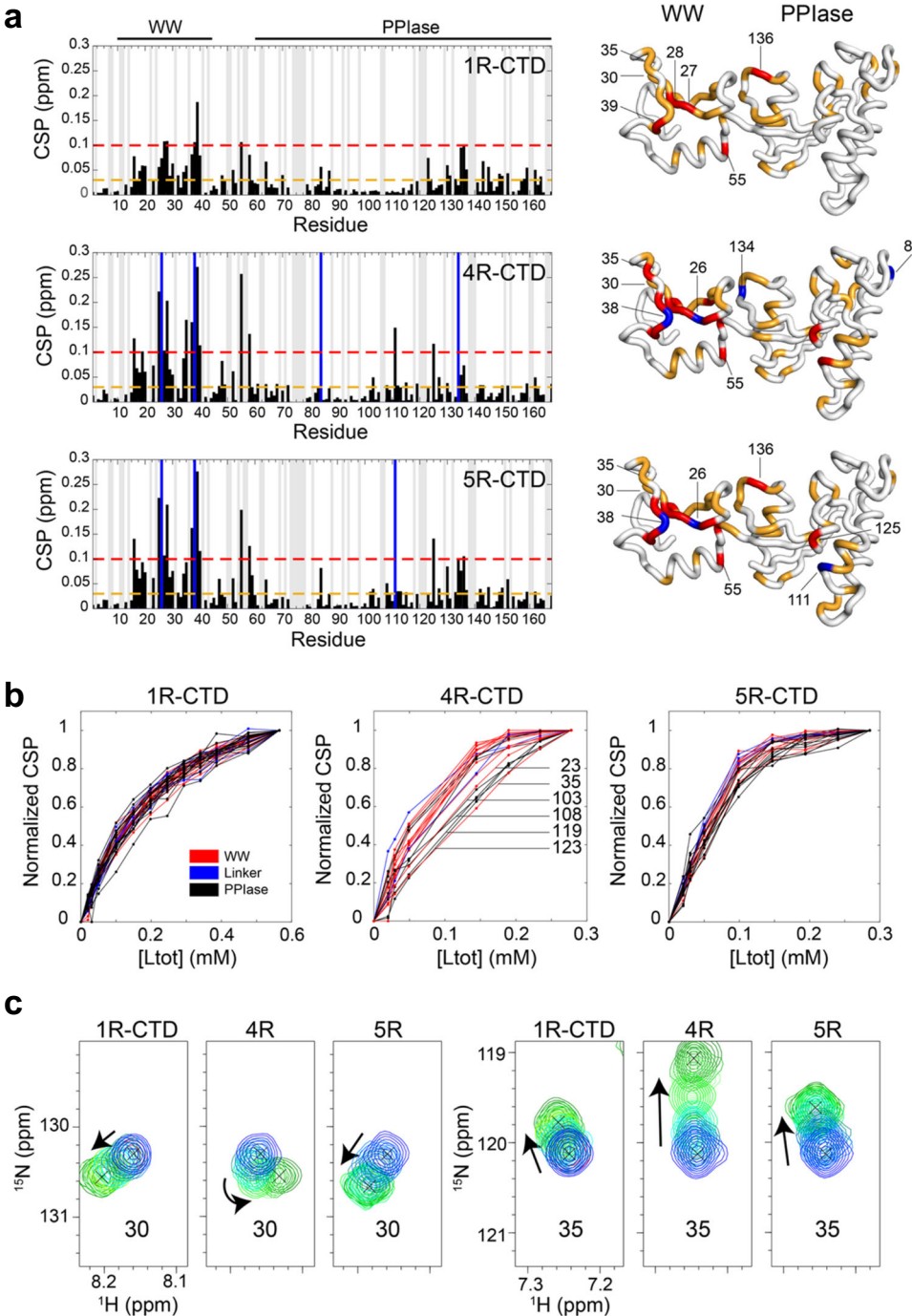

**Fig. 5 NMR chemical shift data reveal simultaneous binding on long CTD peptides. a** $^1$H–$^{15}$N chemical shift perturbations (CSPs) in Ess1 upon binding 1R, 4R, or 5R-CTD peptide at 2:1 peptide:protein stoichiometric ratios (4:1 for 1R-CTD). Since the 1R-CTD has only one binding site, whereas 4R and 5R each have two binding sites, we plot 1R-CTD CSPs at double the stoichiometric ratio as the other CTDs. (Right) CSPs are mapped onto the structure of Ess1, and color-coded orange and red for CSPs > 0.03 and 0.1 ppm, respectively. Residues highlighted in blue are those amide resonances that were broadened beyond detection. Gray bars indicate those resonances for which assignments were ambiguous and therefore not analyzed for CSP plots. **b** Normalized titration curves of Ess1 residues as unlabeled CTD peptide was added (Ltot = total concentration of ligand). Residues are color-coded indicating where the residue is located in Ess1: red (WW), blue (linker), and black (PPIase). **c** Chemical shift trajectories for WW residues 30 and 35 following titration with 1R, 4R, and 5R CTD peptides (see the "Methods" section).

4R-CTD might span that distance (Fig. 6b). This is supported by the directionality of the CTD peptides: the C-terminus of the 2R-CTD emerges from the WW domain near the N-terminus of a 2R-CTD in the PPIase domain.

We also constructed a structural model of how a 5R-CTD peptide could interface with both the WW and PPIase domains

(Fig. 6c). The modeling implies that while a 4R peptide could simultaneously occupy both sites, a 5R peptide would do so without requiring any conformational change in Ess1. Importantly, the orientation and positioning of the 5R peptide in the PPIase domain of Ess1 in our model is fully consistent with an NMR study of the Pin1 catalytic mechanism[35], which indicated

WW domain          PPIase domain

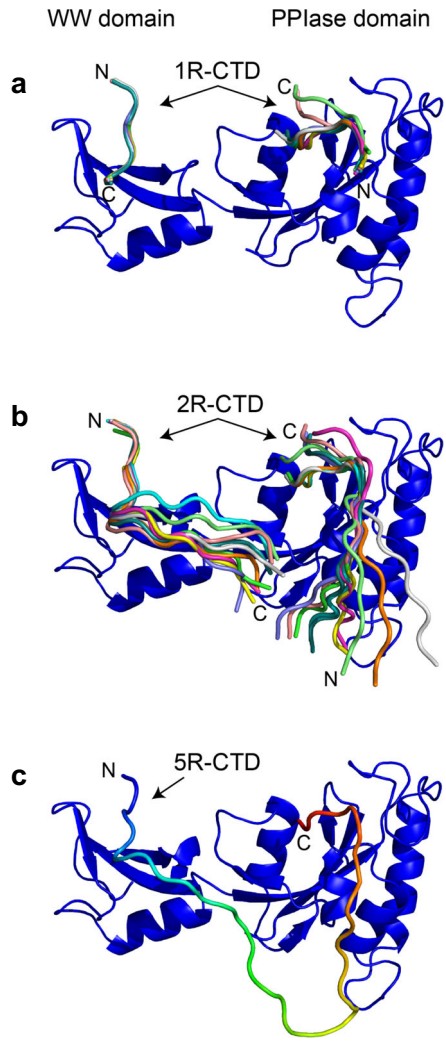

**Fig. 6 Model of Ess1 interactions with CTD peptides. a** 1R-CTD or 2R-CTD was docked to the WW domain or the PPIase domain of Ess1 using PDB 1F8A or PDB 3TCZ, respectively (see text). Top 10 docking structures were superpositioned with peptide direction denoted by labels for N and C-termini. **b** Model of simultaneous binding of two 2R-CTD peptides to the WW and Ess1 domains. Note that the C-terminus of the 2R-CTD bound to the WW domain is near the N-terminus of 2R-CTD bound to the PPIase domain. **c** Model of 5R-CTD peptide interacting with Ess1 using docking results from parts **a** and **b**.

that during *cis/trans* isomerization, the proline and residues C-terminal to it are held in a fixed position deep within the active site, while the upstream, N-terminal portion extends outwards toward the basic loop of the enzyme and rotates 180° during isomerization. To further illustrate this point, we generated a structural model of the 5R-CTD such that the C-terminal pSer-Pro motif was in the *cis* conformation (see the "Methods" section). This *cis* model in conjunction with the *trans* model shown in Fig. 6c highlight the ability of Ess1 domains to simultaneously interact with non-adjacent CTD repeats and catalyze *cis/trans* proline isomerization (Supplementary Movie 1).

Finally, we overlaid Ess1 CSPs obtained from 5R-CTD titration experiments onto the structural model (Fig. 7). Remarkably, the CSPs map out a putative binding interface between the 5R-CTD and Ess1, including residues D103, E111, A112, K115, and R125 in the PPIase domain. Notably, CSPs for these residues only appeared when 4R or 5R CTD was titrated, but not with shorter length CTDs (Fig. 5a). Importantly, the docking results did not

use experimental NMR data as structural constraints. Thus, our NMR data, in conjunction with SV-AUC and FA results, suggest that the Ess1 WW and PPIase domains bind simultaneously to a long, physiologically relevant substrate. We propose that the 5R-CTD is the minimal length for optimal, simultaneous binding to Ess1. That a dual binding mode would increase the overall affinity of Ess1 interaction with a 5R-CTD peptide is consistent with studies using artificial bivalent substrates for Pin1[47] although in the case of Pin1, that length was much shorter (~9 residues between Pin1-binding sites, vs. 28 residues between Ess1-binding sites in the 5R peptide).

## Discussion

**Structural and functional differences between Ess1 and Pin1.** The structure of the *S. cerevisiae* Ess1 reveals conserved folds for the WW domains and PPIase domains, consistent with the fact that orthologs ranging from *C. albicans* Ess1 to human Pin1 complement *ess1* deletion mutants in *S. cerevisiae*. However, the elongated structure of Ess1 and distinct linker region raises a number of important questions. How does the more rigid structure of the fungal enzymes and distinct juxtaposition of the two protein domains influence (or restrict) substrate interactions? Put another way, why does the mammalian Pin1 enzyme lack a highly structured linker found in the fungal Ess1 enzymes, and what possible evolutionary advantage might that confer? Finally, what is the role of the prominent linker α-helix found in the fungal enzymes?

We suggest that the interdomain flexibility of the mammalian orthologs of Ess1 increases the diversity of substrates that can be recognized using a concerted simultaneous binding mechanism. Indeed, human Pin1 is thought to recognize hundreds of potential targets, while multiple genetic studies in yeast have only revealed a limited number of targets[8,48,49]. For more rigid proteins like Ess1 and CaEss1, the flexibility required for simultaneous binding may instead reside in the substrates themselves, for example, in long polymeric targets like the CTD, whose pSer-Pro-binding motifs are less spatially constrained than in globular proteins. The potential differences in substrate preferences makes the explicit prediction that, unlike the ability of Pin1 to complement in yeast, the fungal enzymes would not be capable of fully substituting for Pin1 in mammals. Finally, the prominent solvent-exposed α-helix found in the Ess1 and CaEss1 enzymes could mediate fungal-specific protein–protein interactions.

A model for interdomain communication has been proposed for Pin1[50], whereby the WW domain, upon binding substrate, transmits an allosteric signal via a hydrophobic interface to the PPIase domain, inhibiting its catalytic activity. Not all studies support this model[24,39,51]. For both the *S. cerevisiae* and *C. albicans* Ess1s, this mechanism is not likely because the positioning of the domains is quite distinct, the proposed interface is absent, and many of the key residues proposed to mediate this allostery (Pin1 I28, N30, S138, A140) are not conserved (Ess1 P31, K34, A145, Q147). Instead, we suggest that the fungal Ess1 enzymes are "constitutively active," and only the mammalian Pin1 enzymes may be subject to this regulation.

**Dual binding mechanisms for targeting the CTD.** The long length of the CTD in organisms ranging from yeast (26 repeats) to humans (52 repeats) is likely to enable simultaneous occupancy of distinct protein co-factors to the transcribing RNAPII complex to promote transcription and RNA processing[16,52,53]. The repeated nature of the CTD also provides the opportunity for proteins with multiple CTD-binding domains to interact simultaneously with multiple repeats of the CTD. This has been observed for

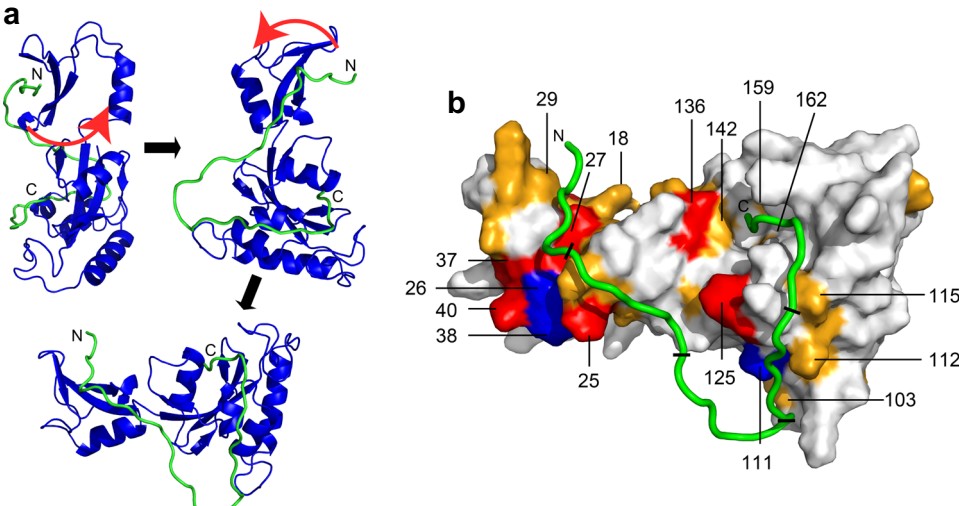

**Fig. 7 Model of Ess1 binding to a 5R-CTD peptide. a** This panel shows the orientation of Ess1 in this figure relative to that in Fig. 1a. The structure of Ess1 is from Fig. 1a, and the peptide model is from Fig. 6c. The N- and C-termini of the peptides are marked as shown. **b** CSPs are mapped onto the space-filling structure of Ess1, and color-coded orange and red for CSPs > 0.03 ppm and CSPs > 0.1 ppm, respectively. Residues highlighted in blue are those amide resonances that were broadened beyond detection. The black slash marks demarcate the five individual CTD heptad repeat units. Note the C-terminal portion of the peptide containing the pSer-Pro motif lies deep in the active site. The residues N-terminal to the peptidyl–prolyl bond in the active site would be able to rotate 180° as suggested by NMR studies with Pin1 (see text and Supplementary Movie 1).

*C. albicans* capping enzyme, Cgt1, which binds to non-adjacent heptad repeats in a pSer5-phosphorylated CTD peptide, effectively looping out an intervening heptad[54]. The yeast termination factor, Nrd1, also binds multiple CTD repeats, and in this case binding to the first of two repeats requires a pSer5-Pro6 motif in the *cis* conformation[55]. The human negative elongation factor PHF3 protein, related to yeast Bye1 (a suppressor of Ess1)[56], uses a newly identified SPOC domain to simultaneously engage two adjacent CTD repeats[57]. The yeast RNA processing enzymes, Pcf11 and Rtt103 bind cooperatively (as homodimers) to long pSer2-phosphorylated CTD peptides (4R), but not to short CTD peptides (2R) with the Rtt103 showing a higher degree of cooperativity[58]. These and other examples provide evidence that the repetitive nature of the CTD is utilized for simultaneous binding via multi-domain and multimeric protein interactions.

Here, we have presented data consistent with a model (Fig. 7) in which the Ess1 WW and PPIase domains bind simultaneously to non-adjacent CTD repeats. We suggest that the length of a 5-repeat CTD peptide would (i) allow simultaneous binding without conformational strain on the Ess1 protein and (ii) provide a sufficient substrate length and flexibility to allow a 180° rotation around the pSer-Pro bond while the N-terminus of the peptide remains anchored to the WW domain (Supplementary Movie 1). Isomerization of a shorter substrate (e.g. 4R) would generate strain on both the CTD and isomerase that would reduce overall affinity. The data also imply that in vivo, Ess1 (and potentially Pin1) could simultaneously engage distal sites within the 26-repeat CTD present in yeast (or 52 in human), generating loops that might sequester CTD-binding proteins, or generate intermolecular bridges between CTDs from distinct RNAPII Rpb1 subunits that might influence RNAPII condensation.

## Methods

**CTD peptides**. The 1R, 2R, 3R, 4R, and 5R CTD peptides were purchased, synthesized with phosphorylation on specific serine residue(s), and HPLC purified to >90% by ABclonal (Table 2). NMR peptides were capped with N-acetyl and C-amide functional groups. FITC peptides were C-amidated. Peptides were resuspended in buffer (20 mM NaPhos, 150 mM NaCl, 3 mM TCEP, 0.02% NaN₃, pH 6.8). Peptide concentrations were determined by measuring $A_{280}$ values using a Nanodrop ND-1000 spectrophotometer and using sequence-determined molar extinction coefficients (e.g. 1280 $M^{-1}$ $cm^{-1}$ for each Tyr residue).

**Protein expression and purification**. *Saccharomyces cerevisiae* Ess1 (originally from strain DBY864 with a previously observed R8S polymorphism)[59] was sub-cloned from a pET28a-ESS1 expression plasmid[18] by PCR with the addition of NcoI and EcoRI for ligation into the pHis.parallel vector (a pET22b derivative)[60]. The resulting plasmid (pKN02) was transformed into Rosetta BL21 pLysS cells. Plasmids were maintained on plates and in liquid media using carbenicillin (50 µg/mL) and chloramphenicol (20 µg/mL). Single colonies were used to inoculate 50 mL Terrific Broth II (TBII, MP Biomedicals) starter cultures that were grown overnight at 30 °C with 200 rpm shaking. 10 mL was used to inoculate 1 L of TBII in baffled expression flasks. The cells were grown at 37 °C for 3–4 h with 200 rpm shaking until the OD₆₀₀ was ~1.0. The flasks were cooled at 4 °C for 1 h, then Isopropyl β-D-thiogalactopyranoside was added to a final concentration of 1 mM. and the flasks moved to 16 °C with 200 rpm shaking for a minimum of 16 h for protein induction. Post-induction, the cultures were spun for 30 min at 4000 rpm (4 °C) and the pellet resuspended with 25 mL of TBII for storage at −80 °C. Frozen pellets were thawed by sitting in RT ddH₂O for ~1 h, then resuspended with 50 mL of lysis buffer (5 mM Tris, pH 8.0; 500 mM NaCl; 20 mM Imidazole; 1 mM DTT; 50 µL 0.1 M phenylmethylsulfonylfluoride (PMSF) and 1 complete protease inhibitor tablet (Roche). Resuspended pellets were lysed with a microfluidizer (Model M-110L, Microfluidics Intl.) and the lysates were cleared by centrifuging for 30 min at 17,000 rpm at 4 °C (JA-20 rotor, Beckman). The crude extract was diluted to 250 mL with Column buffer (5 mM Tris, pH 8.0; 500 mM NaCl; 20 mM Imidazole; 1 mM DTT) and then loaded on a 5 mL HisTrap column (GE Healthcare) on an AKTA purifier at 0.5 mL/min overnight. The protein was eluted with a 25CV, 0–100% linear gradient of Elution Buffer (5 mM Tris, pH 8.0; 500 mM NaCl; 500 mM Imidazole; 1 mM DTT) and fractionated into 2 mL fractions. Fractions were checked for protein using a 4–15% polyacrylamide gel (BioRad) and then combined. GST-TEV (produced in our lab) was then added to the pooled fractions for 6xHis-tag cleavage. The sample was placed in a 6–8000 Da dialysis membrane and floated in 1 L of Column Buffer for imidazole removal for a minimum of 6 h. The buffer was changed twice prior to collecting the sample, then it was put over the same HisTrap column to remove any proteins bound non-specifically. After pooling, the sample was concentrated to <5 mL and loaded on a 20/200 HL Sephadex Gel Filtration column with Gel Filtration Buffer (20 mM Tris, pH 8.0; 250 mM NaCl; 5 mM DTT) as a final purification step. The protein was concentrated to a minimum of 1 mM and frozen in 50 µL aliquots in liquid nitrogen. These were stored at −80 °C until used. Alternatively, the protein was also gel-filtration-purified into a buffer better suited for AUC (20 mM Tris, pH 8.0; 250 mM NaCl; 2 mM TCEP) and then frozen in a similar fashion.

**Protein crystallization and data collection**. Purified ScEss1 was screened against the JCSG core I suite crystallization screen (Hampton research) using hanging drop crystallization method. Screening was performed using the protein alone, as well as in the presence of a 1-repeat CTD peptide[48], which was biotinylated on the N-terminus and contained additional flanking residues (biotin-GGSGGS-**YSPTpSPS**-YS). Condition 14 of this screen (0.1 M HEPES, pH 7.5; 20% (w/v) PEG 8000) gave an initial hit, which was then optimized. The condition that provided the crystal from which the structure was determined was 0.1 M Tris, pH 7.7 and 21% (w/v) PEG 8000.

The protein concentration was 30 mg/mL with an equimolar concentration of 1R-CTD peptide and was mixed at a 2:1 ratio of protein:reservoir. The crystal was frozen in liquid nitrogen in a modified crystallization condition (0.1 M HEPES, pH 7.5; 18% PEG 8000; 15% (v/v) glycerol) and stored for ~2 months prior to collecting data. Diffraction data were collected on the F1 Beamline ($\lambda = 0.917$ Å, $T = 100$ K) at the Cornell High Energy Synchrotron Source (MacCHESS). Diffraction data were obtained with 3 s exposures and 1° rotations around Phi for an entire 360°. The space group determined for the crystal was C2. The data were indexed, reduced, and scaled using the HKL-2000 software package in the Harvard SBGrid consortium[61]. Initial phases were obtained by molecular replacement with Phaser: the search model used was the PPIase domain of the *C. albicans* Ess1 structure[34] (PDB ID: 1YW5). After an initial rigid body refinement, auto-building was performed using ARP/WARP[62]. Standard structural modeling and refinement were performed with Coot[63] and PHENIX[64], respectively. Ramachandran statistics: favored (94.5%), allowed (4.5%), and disallowed (1.0%). The first eight residues of Ess1, which are N-terminal to the WW domain, were not well ordered. The asymmetric unit consisted of two Ess1 proteins, one of which was complete and the other lacked density for residues 1–42, which includes most of the WW domain.

**Biological small-angle X-ray scattering (SAXS) analysis.** Purified ScEss1 was thawed and run over the S200 size-exclusion column again to buffer-exchange it into SAXS buffer (20 mM Tris, pH 8.0; 250 mM NaCl; 5 mM DTT; 3% (v/v) glycerol). The protein was analyzed on the CHESS G1 beam line (Ithaca, NY) using a 250-µm square x-ray beam with a flux of $\sim 3 \times 10^{11}$ photons/s/mm$^2$ at 9.96 keV. Measurements were made at a wavelength of 1.244 Å and at 4 °C using a dual Pilatus 100K-S detector. Ten 1–2 s exposures of each 30 µL sample were obtained, with sample oscillation to help prevent radiation damage. Three different protein concentrations were tested (4.78, 3.19, 1.57 mg/mL) and each provided good-quality data. Initial processing, including frame averaging and buffer subtraction, was done using the RAW software. The SAXS scattering data were plotted as Guinier curves at increasing concentrations, which for all datasets showed linearity in the low $q$ angles, indicating that the samples were free of aggregation, radiation damage or interparticle effects over the concentration range. Guinier approximation was applied to low $q$ scattering region, and the radius of gyration ($R_g$) was determined from a linear fit to the Guinier plot ($\ln(I)$ vs. $q2$) for the $q$ range that satisfies the relationship $qR_g < 1.3$ in the program Primus (ATSAS Package, EMBL). Plots of the $R_g$ as a function of concentration were linear with minimal concentration dependence (slope = 0.31) and extrapolation to infinite dilution gave an $R_g$ of 19.8 Å.

The pair distance distribution function ($P(r)$) was calculated using the indirect Fourier transform method in the program GNOM (ATSAS Package, EMBL). The maximum protein dimension ($D_{max}$) values were determined from the $P(r)$ analysis, where $P(r)$ approaches zero. Low-resolution ab initio envelopes were calculated from the GNOM program outputs with a high-resolution limit such that $q_{max} \leq 8/R_g$ using the program DAMMIF (ATSAS Package, EMBL). Ten individual models were calculated. The program DAMAVER (ATSAS Package, EMBL) was used to align the 10 models. The aligned models were further refined using the program DAMMIN (ATSAS Package, EMBL). Three-dimensional ab initio molecular envelopes (total of 10) were calculated from the SAXS data, which showed an average normalized spatial discrepancy value of ~0.55, indicating that each of the models were highly similar. Theroretical scattering profiles derived from crystal structures were calculated using Primus program Crysol (ATSAS Package, EMBL).

**Analytical ultracentrifugation and isotherm analysis.** Purified ScEss1 (10 nM–1 mM), either with or without denatured-length, FITC-labeled CTD peptides (50 µM), was loaded into an AUC cell with a 3 or 12 mm Epon charcoal centerpiece (SpinAnalytical) sandwiched between sapphire windows. The cells were loaded into a Ti-60 titanium rotor pre-equilibrated to 10 °C in a Beckman-Coulter XL-A analytical ultracentrifuge. Chamber vacuum was re-established and temperature re-equilibration was performed for a minimum of 2 h. A wavelength scan at 3000 rpm from 200–350 nm was then run to test that the absorbance levels were within the range of 0.1–1.0 OD. Conversely, the long-pass filter would be moved to the horizontal position prior to re-equilibration and the wavelength scan would be performed at the same speed but with a window of 400–550 nm for FITC peptide absorbance. The selected wavelength (normally 280 or 490 nm, respectively) was then used for the method scan, which was as follows: 60,000 rpm, 300 scans per cell with 0 s delay between scans at 10 °C. Data were analyzed using SEDFIT and $c(s)$ distributions were then loaded into GUSSI (www.utsouthwestern.edu/labs/mbr/software/) for isotherm-building. Distributions were integrated between 0.5S and 4S and the resultant $s_w$ isotherm was then analyzed by simulation with explicit entry steps for binding to and dissociation from each binding site. Association rate constants were fixed at the value of 1 and the dissociation rate constants were floated to obtain estimate of $K_d = k_{off}/k_{on}$. Dissociation constants for each site were linked and fit by constraining the thermodynamic cycle to enforce conservation of energy using the following equation for the output observable:

$$\text{Two site binding model}: \theta = a + (b*((ES + SE + 2*SES)/(E + ES + SE + SES))) \quad (1)$$

where $a$ is the minimum $s_w$ value and $b$ is the maximum $s_w$ value range. $E$ is the enzyme and $S$ is the ligand binding to either site one ($SE$) or site two ($ES$).

Values for $a$ (0.61 ± 0.07) and $b$ (1.48 ± 0.18) were determined from analysis of the 5R-CTD titration and fixed at those values for the initial analysis of all isotherms. For the 4R isotherm, to account for the possibility of a conformational change that alters the $s_w$ values, the maximum $s_w$ range ($b$) was allowed to float, fitting with a value of 1.04 ± 0.02.

**Fluorescence anisotropy.** Fluorescence anisotropy measurements were performed in a Hidex Sense microplate reader using an excitation wavelength of 485 nm and an emission wavelength of 535 nm. Measurements were performed in 20 mM Tris pH 8.0, 3 mM TCEP buffer. Purified wild-type ScEss1 was titrated into a constant 1 µM FITC-labeled 1R-CTD peptide to determine binding affinity of Ess1 to 1R-CTD by anisotropy measurements. Addition of Ess1 protein or non-fluorescent CTD peptides did not affect the measured fluorescence intensity of the FITC-CTD peptides. For the fluorescent competition experiments, unlabeled CTD peptides (1, 2, 3, 4, or 5 repeat peptides) were titrated (final concentrations ranging from 0 to 350 µM) into a solution containing a constant 50 µM Ess1 and 1 µM FITC-1R-CTD peptide. Competition data was used to calculate $IC_{50}$ values to compare the affinity differences between all five substrates (Eq. (1)).

$$\text{Competition model}: f1 = \min + (\max - \min)/(1 + (x/IC_{50}) \wedge (-\text{Hillslope})) \quad (1)$$

**$^{13}$C/$^{15}$N Ess1 C120S expression and purification.** Full-length Ess1 or Ess1 C120S plasmid was used to transform *E. coli* cells (Rosetta II (DE3) pLysS; Novagen) with Chloramphenicol and Carbenicillin selection. Single colonies were selected and grown in 1 L terrific broth (TB II) media at 37 °C overnight (16 h). Cells were harvested by centrifugation at 4000×$g$ for 30 min at 4 °C to remove all carbon and nitrogen sources, then re-suspended in 25 mL M9 minimal media, then transferred to a flask containing the remainder of the 1 L M9 media for growth and expression (50 mM Na$_2$HPO$_4$, 50 mM KH$_2$PO$_4$, 5 mM Na$_2$SO$_4$, 2 mM MgSO$_4$·7H$_2$O, pH 7.4) supplemented with antibiotics, 3 g/L $^{15}$N-ammonium chloride and 4 g/L $^{13}$C-D-glucose (Cambridge Isotope Laboratories, Inc.). Minimal media cultures were grown at 37 °C in a shaking incubator at 200RPM until they reached an OD$_{600}$ of ~0.6, and expression was induced with 1 mM isopropyl β-D-1-thiogalactopyranoside (IPTG). Purification of protein was similar as described above. As a final step of purification, proteins were passed through a HiLoad 16/60 S200 size exclusion column (GE Healthcare) that was pre-equilibrated with buffer (20 mM Tris, 150 mM NaCl, 3 mM Tris(2-caboxyethyl) phosphine (TCEP), pH 8.0). Pooled fractions were concentrated to 1.4 mM using a 10 kDa MWCO spin concentrator (Millipore) and dialyzed into pH 6.8 buffer containing 9.1 mM Na$_2$HPO$_4$, 10.7 mM NaH$_2$PO$_4$, 150 mM NaCl, 3 mM TCEP, 0.02% w/v NaN$_3$, resulting in a final concentration of 500 µM $^{15}$N,$^{13}$C-Ess1 which was stored at −80 °C prior to NMR experiments.

**NMR instrumentation.** NMR experiments were performed at 298 K on a Bruker Avance III HD 800 MHz spectrometer, equipped with TCI Cryo Probe. All NMR data were processed using NMRPipe[65] and analyzed using CCPNMR 2.4.2[66].

**Resonance assignments for Ess1.** To obtain chemical shift assignments, NMR samples consisting of 500 µM $^{15}$N,$^{13}$C-labeled Ess1 C120S were prepared in pH 6.8 buffer with 20 mM NaPhosphate, 150 mM NaCl, 3 mM TCEP, 0.02% w/v NaN$_3$, and 5% D$_2$O. All experiments were obtained at 298 K. Chemical shift assignments of the backbone resonances (H$^N$, N, CO, Cα, and Cβ) were obtained using 2D $^1$H–$^{15}$N, HSQC, and standard triple resonance experiments (HNCO, HNCACO, HNCACB, and CBCACONH) spectra. Acquisition times for the triple resonance experiments were 92 ms in the direct $^1$H dimensions, 17–19 ms in the indirect $^{15}$N dimensions, 16–20 ms in the indirect $^{13}$CO dimensions and 6 ms in the indirect Cα/Cβ dimensions. Spectral widths were 12 ppm in $^{13}$CO, 65 ppm in $^{13}$Cα/Cβ dimension, 26 ppm in indirect $^{15}$N. Non-uniform sampling (NUS) was employed for all triple resonance experiments. Experiments were acquired with 20–25% (CACBCONH 20%; HNCACO, HNCO, and HNCACB 25%) sampling using the Poisson Gap sampling method[67] NUS spectra were processed using SMILE and NMRPipe[65,68] and employed standard apodization parameters and linear prediction in the indirect dimensions. Using these experiments, we successfully assigned amide backbone resonances (H$^N$, N) for 92% of all residues. For $^{13}$C chemical shifts, 95% (154/162) Cα, 93% (141/151) Cβ, 88% (143/162) CO were assigned.

**Wild-type Ess1 NMR spectroscopy.** All relaxation or titration NMR experiments using CTD peptides were collected using wild-type (WT) Ess1, not the C120S mutant described above. For this reason, we transferred backbone $^{15}$N and $^1$H chemical shift assignments to WT. We prepared a wild-type (WT) NMR sample of scESS1 using 430 µM $^{15}$N labeled WT Ess1 in 20 mM NaPhos pH 6.8, 150 mM NaCl, 0.02% NaN$_3$, 3 mM TCEP, 5% D$_2$O (the same conditions used for chemical shift assignment of Ess1 C120S). 2D $^1$H–$^{15}$N HSQC spectra were collected, and the vast majority (151/153) of the backbone amide assignments were transferred to WT from the C120S Ess1 spectra. ~22% (~34 out of 153) of $^{15}$N–$^1$H amide backbone assignments were shifted slightly, and these were for residues spatially near C120S (Supplementary Fig. 9).

**NMR relaxation measurements**. Backbone amide $^{15}$N $R_1$ and $R_2$ relaxation rates and heteronuclear $^1$H–$^{15}$N NOE were measured for $^{15}$N-labeled WT Ess1 using 235 µM protein in pH 6.8 buffer consisting of 20 mM NaPhosphate, 150 mM NaCl, 3 mM TCEP and 0.02% NaN$_3$. We used established pseudo-3D interleaved relaxation pulse sequences and protocols[69]. The following time delays were used for $R_1$ experiments: 4 ms (×2), 1000 ms (×2), and 1600 ms (x 2). For $R_2$ experiments, we used time delays of 8 ms (×2), 24, 32, 48 ms (×2), 64 ms (×2), and 88 ms. The heteronuclear $^1$H–$^{15}$N NOE experiments were collected with an interscan delay of 5 s. For $^{15}$N $R_1$ and $R_2$ relaxation experiments, the $^1$H and $^{15}$N acquisition times were 100 and 29 ms, with spectral widths of 12 and 24 ppm for the $^1$H and $^{15}$N dimensions, respectively. For heteronuclear $^1$H–$^{15}$N NOE experiments, the $^1$H and $^{15}$N acquisition times were 100 and 29 ms, with spectral widths of 12 and 28 ppm for the $^1$H and $^{15}$N dimensions, respectively. Relaxation rates were determining using RELAXFIT[70]. The overall rotational diffusion tensors for Ess1 and individual WW and PPIase domains were determined using ROTDIF[71,72] and the crystal structure of Ess1 described here. Residues only in well-defined secondary structure elements were used for ROTDIF calculations. The ratio of relaxation rates was determined for each residue as $\rho = (2R_2'/R_1'-1)^{-1}$, where $R_1'$ and $R_2'$ are modified $^{15}$N $R_1$ and $R_2$ rates with the high-frequency components subtracted[73]. Errors in $^{15}$N $R_1$ and $R_2$ relaxation rates were determined using 500 Monte Carlo trials in RELAXFIT. Errors in heteronuclear $^1$H–$^{15}$N NOE measurements were determined using the standard error propagation formula.

**CTD peptide titration experiments with Ess1**. Mixed NMR samples of Ess1 and CTD peptides (1R, 2R, 3R, 4R, or 5R CTD) consisted of 100 µM $^{15}$N WT Ess1 and 50–200 µM CTD peptide. Samples were mixed at 1:2, 1:1, and 2:1 protein to peptide ratios in pH 8 buffer containing 20 mM Tris, 0.02% NaN$_3$, 3 mM TCEP, and 5% D$_2$O. A series of 2D $^1$H–$^{15}$N HSQC spectra were collected at the different protein to peptide ratios, and chemical shifts were reassigned by visual inspection as titration experiments were conducted in a different buffer than used for chemical shift assignments (see above). HSQC experiments were collected with acquisition times of 90 and 36 ms in the $^1$H and $^{15}$N dimensions, respectively. Spectral widths were 14 and 34 ppm for $^1$H and $^{15}$N, respectively.

**Ess1-CTD peptide NMR $K_d$ determination experiments**. 1R, 4R, and 5R CTD peptides used in the $K_d$ titration experiments were the same as above. The peptides were titrated from concentrated stocks (1.0–7.6 mM) into 100 µM samples of WT $^{15}$N Ess1, with peptide to protein ratios ranging from 0 to 6 (for 1R) and 0 to 3 (for 4R and 5R-CTD). Binding was monitored by recording $^1$H–$^{15}$N HSQC spectra at different peptide to protein ratios. For each backbone amide, we assumed that the CSP was a weighted average between the free ($\Delta\delta = 0$) and ligand-bound ($\Delta\delta = \Delta\delta_{bound}$) states, such that $\Delta\delta = \Delta\delta_{bound} * [PL]/[P_{total}]$. Here, [PL] and $P_{total}$ represent the ligand-bound protein concentration, and $P_{total}$ is the total protein concentration of Ess1. The CSP for each amide was calculated using the equation $\mathrm{CSP} = \sqrt{(\Delta\delta_H)^2 + \left(\frac{\Delta\delta_N}{5}\right)^2}$ where $\Delta\delta_H$ and $\Delta\delta_N$ represent the change in chemical shift in the $^1$H and $^{15}$N dimensions, such that $\Delta\delta = 0$ when calculating the CSP for the reference spectrum (in the absence of ligand). To obtain the dissociation binding constant, $K_d$, we used the single-site binding equation, $[PL] = (([P_{total}] + [L_{total}] + K_d) - \sqrt{([P_{total}] + [L_{total}] + K_d)^2 - 4 * [P_{total}] * [L_{total}]})/2$, where $[P_{total}]$ and $[L_{total}]$ are total protein (Ess1) and ligand (CTD peptide) concentrations, respectively. Fitting was performed using an in-house Matlab script. Overall $K_d$ was determined by averaging residue-specific $K_d$ values for those residues with CSP > 0.03 ppm at the titration endpoint (Table S3). Errors represent standard deviation of residue-specific $K_d$ values.

**Mass spectrometry**. The molecular weights of the 1R, 4R, and 5R CTD peptides (used for NMR titration experiments, see Table 2) were verified using electrospray ionization MS (ESI-MS) (Fig. S10). Peptides were cleaned up using a C18 solid phase extraction cartridge. For positive or negative mode ESI-MS, 10 µM peptide in 50% acetonitrile, 0.1% formic acid was injected at 10 µL/min into an Orbitrap LUMOS mass spectrometer.

**Molecular docking and 5R-CTD Ess1 complex modeling**. Starting models of Ess1 bound to CTD peptides were generated as described below, and refined using the program, Rosetta-FlexPepDock[74]. Approximately 200–300 high-resolution structures were generated from starting models that were submitted to the Rosetta-FlexPepDock webserver (http://flexpepdock.furmanlab.cs.huji.ac.il/index.php). No additional constraints were used. All prolines were in *trans* conformation unless otherwise noted below. The starting model of Ess1 WW domain bound to 1R-CTD was based on the crystal structure of Pin1 WW–CTD complex[23]. The Pin1 crystal structure (PDB code 1F8A) was aligned to the structure of Ess1, with the two WW domains aligning with a Cα RMSD of 0.553 Å. The extra phosphate group on Ser-2 of the CTD peptide from Pin1 crystal structure was manually removed. The starting model of Ess1 PPIase domain bound to 1R-CTD was prepared using the crystal structures of Pin1 PPIase domain bound to a peptidomimetic inhibitor in the active site (PDB codes 3TCZ and 3TDB)[19]. The inhibitors in these crystal structures mimic the *cis* and *trans* conformers of the phosphoserine-proline (pSP)

motif in the CTD peptide. Using Pymol 2.0, a 1R-CTD peptide (YSPTpSPS) was built and its pSP motif was structurally aligned to the pSP section of the *trans* inhibitor (PDB code 3TDB). (The *cis* inhibitor has similar local structure). The rest of the peptide was manually adjusted to avoid steric clashes with the protein, then the complex was optimized using Rosetta-FlexPepDock. Models of Ess1 bound to 2R-CTD peptides were generated using 1R-CTD optimized structures. To generate the starting model of 2R-CTD bound to the WW domain of Ess1, another repeat of CTD peptide not containing a pSP motif (YSPTSPS) was built onto the C-terminus of the existing peptide using the build function in Pymol. To generate the starting model of 2R-CTD bound to the PPIase domain of Ess1, another repeat of CTD peptide (YSPTSPS) was built onto the N-terminus of the existing peptide. From each of the 2R-CTD Rosetta-FlexPepDock results for the WW and PPIase domains, one structure was selected to be representative of the top 10 structures. Using Pymol's build function, one additional CTD repeat (YSPTSPS) was added and shaped to connect the two 2R-CTDs to form a complete 5R-CTD peptide spanning across both the WW and PPIase domains.

A separate 5R-CTD model with *cis* conformer of the pSP motif in the PPIase domain active site was built using the same method described above, except that we used the pSP section of the *cis* inhibitor (PDB code 3TCZ). We then used the "morph" function in Pymol to generate a movie of how the pSP motif in the 5R-CTD peptide could be isomerized from *trans* to *cis* states (Supplementary Movie 1).

**Statistics and reproducibility**. All experients were repeated at least twice, or as indicated in the text. Sample sizes were determined using current standards in the field and from prior experience. No data were excluded. Standard statistical tests were used and are described in the figure legends and respective "Methods" sections.

**PDB codes**. Figure 1 (CaEss1 1YW5; Pin1 1PIN), Fig. 6 (Pin1 WW domain 1F8A, Pin1 catalytic domain 3TCZ), Figs. S2a (peptide from Pin1 1F8A), S2b (CaEss1 1YW5; Pin1 1PIN), S2c (Pin1 1PIN1).

**Reporting summary**. Further information on research design is available in the Nature Research Reporting Summary linked to this article.

## Data availability

Coordinates for the *S.c.* Ess1 X-ray crystal structure have been deposited in PDB with the accession code ID 7KKF. NMR chemical shift assignments of the backbone resonances (H$^N$, N, CO, Cα, and Cβ) are deposited in the BMRB database with the accession code ID 50787. Other datasets are available within a ZIP folder in Supplementary Data 1. Any remaining information can be obtained from the corresponding authors upon reasonable request.

## Code availability

All illustrations were generated using Chimera (Figs.1, 2, S2), PyMol (Figs. 5–7, S4, S6, S9, Movie 1), Microsoft Powerpoint, Adobe Illustrator, and Matlab.

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

## Acknowledgements

We are grateful to colleagues Stewart Loh, Joshua Karchin, and Stephen Shinsky for initial NMR results, Thomas Duncan and Alaji Bah for helpful discussions, and Bede Portz for comments on the manuscript. We thank the staff at the Macromolecular Diffraction Facility at the Cornell High Energy Synchrotron Source (MacCHESS) facilities for assistance with X-ray data collection. Data collection with a Bruker 800 MHz NMR magnet was supported by NIH shared instrumentation grant 1S10OD012254. We thank Ebbing De Jong and the SUNY-Upstate Proteomics MS Core facility for assistance and collection of ESI-MS data. The Orbitrap LUMOS MS spectrometer was supported by NIH shared instrumentation grant 1S10OD023617–01A1. This work was supported by grants from the NIH R01-GM123985 to SDH, NSF CAREER 1750462 to CAC, and NIH R01-CA140522 to MSC.

## Author contributions

S.D.H., M.S.C., and C.A.C. designed the study and contributed to writing the manuscript. K.N. and N.A.-V. carried out the crystallography, K.E.W.N. the SAXS, A.J.C. the FA, and T.Z. the NMR experiments. All authors contributed to the interpretation of data, troubleshooting, and preparation of the manuscript.

## Competing interests

S.D.H. is a co-founder of Kathera Bioscience, Inc., and on the Scientfic Advisory Board. M.S.C. serves on the Consultant Advisory Board for Kathera Biocience, Inc. Remaining authors declare no competing interests.
