## [Peer Review File · Communications Biology]

Reviewers' comments:

Reviewer #1 (Remarks to the Author):

Namitz & colleagues investigate the interaction of the yeast ScEss1 prolyl cis/trans isomerase and the CTD peptide of polymerase II with a battery of biophysical approaches (including SAXS, analytical ultracentrifugation and NMR spectroscopy) to study the relationship between the two domains of Ess1.

As a founding member of the essential parvulin type prolyl cis/trans isomerases, of which the human Pin1 through its myriad of putative substrates linked to human disease is the best known, it remains interesting to understand how both domains (WW and catalytic) act together, and especially how they might differentially act together in different organisms. Here, the authors determine that Ess1 is more rigid in terms of inter-domain interaction than is Pin1. The interesting question is thus how this more rigid architecture influences substrate interactions ?

For Pin1, negative allosteric regulation of the catalytic site by interdomain contact has been suggested (Wang et al. Structure 2015). Actually, for a pCDC25 derived peptide ligand, the activity is higher for the isolated PPIase domain than in the presence of the WW domain. This questions the hypothesis that the purpose of the WW domain is the increase of local ligand concentration in order to increase the activity of Pin1. Instead, the WW domain appears to play a more subtle role in activity regulation, possibly in context with ligand-specific allostery. It would be interesting to evaluate the same for Ess1, but catalytic data (cis/trans isomerase activity as measured by EXSY NMR spectroscopy on the different peptide substrates) are missing. My feeling is that these would truly strengthen the story, unless the authors feel that binding rather than cis/trans isomerase is the main activity? Their movie suggests the contrary, so additional data on this main point would be welcome.

An interesting thought is that the more rigid yeast Ess1 can be replaced by the human Pin1 but not the other way around. Is there any experimental evidence for this (in a mouse model or related) ? This is just out of interest.

In Figure 5C, the authors show a non-linear titration behaviour for the R30 cross peak when the 4R peptide is titrated in the solution. Do other residues show the same behaviour ? And as this has been described in great length by P Wright and colleagues, can the authors translate this into a molecular picture for the WW/catalytic domain behaviour ? Notably the linker region should become constrained in their model, and this should show up in the NMR spectra.

Reviewer #2 (Remarks to the Author):

The manuscript by Namitz and colleagues describes the structure of the Ess1 prolyl isomerase (PPIase) from *S.cerevisiae* determined by X-ray crystallography. The authors investigate the flexibility between the N-terminal WW and the C-terminal PPIase domain by small angle X-ray scattering, NMR spectroscopy, and analytical ultra-centrifugation. They analyze the binding affinity of Ess1 for the CTD hepta-repeats of the RNA polymerase II using fluorescence anisotropy, sedimentation velocity aUC, and titration NMR spectroscopy by applying various length for the substrate peptide from 1 to 5 repeats. They find that five repeats are the minimally required optimal length for binding to the two substrate recognition sites of the Ess1 proteins. Using molecular model approaches they provide a model of the flexible CTD covering the distance between the pS5P6 recognition site at the WW domain at the N-terminus and the second pS5P6 recognition site at the PPIase domain at the C-terminal end of the CTD peptide. Ess1-catalysed proline isomerization of the CTD peptide therefore requires a “lag hepta-repeat” length of 4 repeats, as this is the optimal distance between the recognition site and the active site.

This is a very nice and technically very comprehensive study to challenge the molecular characterization and structure determination of a two-domain protein containing two similar peptide recognition sites with an intrinsically unstructured peptide as the CTD, that is built of a repetitive hepta-repeat structure. The authors apply not only X-ray crystallography for the first structure determination of the ScEss1 protein, but also NMR spectroscopy with a full protein assignment and dynamic studies, small angle X-ray scattering, analytical ultracentrifugation, fluorescence anisotropy and molecular modelling.

I have only some minor comments and suggestions before recommending the manuscript for publication.

Comments:

Maybe I got it wrong but the authors say in the abstract that “the Ess1 WW domain anchors the distal end of the CTD substrate during isomerization”. However, Fig. 7 B shows the N-terminus of the 5 hepta-repeat CTD peptide binding to the WW domain whereas the fifth repeat at the C-terminus is at the PPIase domain. So, the proximal part of the CTD interacts for recognition with the WW domain while four repeats further down in the distal part occurs the isomerization. Please clarify.

Along the line: The Movie Graphic on page 3 should indicate the N- and C-terminus of the CTD peptide. I also suggest making the background white and labelling the WW and PPIase domain to better grasp the picture. As only the backbone of the CTD is shown, maybe the backbone of the tyrosine-1 position could be labelled in a different colour to help identifying the five repeats in the recognition sequence.

In the Introduction section, the authors explain the differences in the flexibility of the linker sequence between CaEss1 and human Pin1 which might impact on the engagement with substrates (lines 114-122). When explaining the new ScEss1 structure in the Results, it would be very nice to shown a superimposition of all three full length isomerases ScEss1, CaEss1 and hsPin1 that is made only on the PPIase domain, to show how differently the WW domain assembles relative to the PPIase. This could be either as a small inset in Figure 1 or in the supplement.

In Figure S2A, the electron density identified for proline 6 is shown (and even a little density for the P in pS5) but the model also displays a pS2. According to the text, only one phosphorylation is present

in the peptide. Please change or clarify. Also, N- and C-term of the peptide should be indicated.

I don't understand how the RMSD values for the PPIase and the WW domain were determined. Both values sound very high to me when looking at the Figures 1B and 1C. First, the three chains are really difficult to identify due to the very similar blueish/greenish colors. Second, if this would be only an alignment on the Ca atom position, I suppose the RMSD values would be much smaller – to me, it looks like 0.6-0.8 Ang. Yet, if all side chains are considered in the alignment, then 1.6 and 1.7 might be reasonable. However, then also the entire molecule should be shown with all side chains.

Fig. 3 would very much benefit from showing at the top the 5 peptides used in this analysis. This could be either simplistic just indicating the site of the fluorescent label and the repeats with the phosphorylation site, or in a letter code as in Table 2.

Other Comments:

Abstract, line 50: As the authors speak of *S.cerevisiae* Ess1, they may refer to the 26 CTD hepta-repeats instead of writing “n”.

Line 99: “and 52 repeats in humans (about half consensus)”. Please specify the alterations from the consensus sequence that occur mostly at position 7 from the hepta-repeat in a short comment.

Stoichiometry of protein-peptide (5R) binding (lines 300-307): This question might be most elegantly addressed by isothermal titration experiments. The dissociation constant (about 10 μ M for 5R) should be sufficient to get good signals. However, such analysis is beyond this study as the study is already very comprehensive with a multitude of techniques applied. But the author could more clearly discuss that not only 1 protein could bind 2 peptides (in case of the short 1R or 2R), but also 1:1 binding for 4R and 5R, while even 1 peptide could bind two proteins (5R). I personally consider the last complex formation as very unlikely and I think the data convincingly show 1:1 binding for the long peptides by aUZ.

Remark:

Nothing for this study, but I like to encourage the authors to try co-crystallization of the ScEss1 protein with a 5-mer CTD peptide (first and last repeat with a Ser5 phosphorylation) and possibly even a sixth repeat at the C-term, as there seems to be still some coverage. Maybe ligand soaking would already work. And did the authors test if a K7 residue changes the affinity for the recognized peptide? It would be very good to know how CTD alterations from the consensus sequence influence proline isomerization. Thanks!

Reviewer #1 (Remarks to the Author):

Namitz & colleagues investigate the interaction of the yeast ScEss1 prolyl cis/trans isomerase and the CTD peptide of polymerase II with a battery of biophysical approaches (including SAXS, analytical ultracentrifugation and NMR spectroscopy) to study the relationship between the two domains of Ess1.

As a founding member of the essential parvulin type prolyl cis/trans isomerases, of which the human Pin1 through its myriad of putative substrates linked to human disease is the best known, it remains interesting to understand how both domains (WW and catalytic) act together, and especially how they might differentially act together in different organisms. Here, the authors determine that Ess1 is more rigid in terms of inter-domain interaction than is Pin1. The interesting question is thus how this more rigid architecture influences substrate interactions ?

(1) For Pin1, negative allosteric regulation of the catalytic site by interdomain contact has been suggested (Wang *et al.* Structure 2015). Actually, for a pCDC25 derived peptide ligand, the activity is higher for the isolated PPIase domain than in the presence of the WW domain. This questions the hypothesis that the purpose of the WW domain is the increase of local ligand concentration in order to increase the activity of Pin1. Instead, the WW domain appears to play a more subtle role in activity regulation, possibly in context with ligand-specific allostery. It would be interesting to evaluate the same for Ess1, but catalytic data (cis/trans isomerase activity as measured by EXSY NMR spectroscopy on the different peptide substrates) are missing. My feeling is that these would truly strengthen the story, unless the authors feel that binding rather

Summary

The interesting aspects of our manuscript were identified and the extensive analyses we performed was recognized. We thank the reviewer for their insights, suggestions, and comments.

(1) We are fully aware of the intriguing model of allosteric control of Pin1, whereby contacts by the WW domain negatively influence catalytic domain activity, as we noted in paragraph 3 of our Discussion. This model for Pin1 function was proposed and studied extensively by the Peng laboratory where they find the isolated PPIase catalytic domain has *higher* activity against Cdc25 peptides than the full-length protein (Wilson *et al.*, Biochemistry, 2013; Wang *et al.*, Structure, 2015). While the Peng laboratory and others (*e.g.* Nicholson laboratory, Rogals *et al.* FEBS J 2016) have evidence supporting this model including substitution mutations at key positions and use of additional peptides (IRAK), the model remains controversial. Other laboratories such as those of D. Kern and G. Lippens do not find evidence for WW domain-induced negative allostery and in fact, show that for other Pin1 substrates (*e.g.* *tau*) the isolated PPIase domain has a *lower* activity than full length (Eichner *et al.*, JMB, 2016; Smet *et al.*, FEBS Lett., 2005). Still others have argued that depending on what substrates are used, allostery can be either positive or negative (Zhu *et al.* J. Phys. Chem. Lett, 2019). For our Ess1 study, however, much of this controversy is not relevant. As explained in the Discussion (lines 431-438), the positioning of the Ess1-WW domain would not be compatible with the Peng allostery model, and the residues they identified as being

than cis/trans isomerase is the main activity? Their movie suggests the contrary, so additional data on this main point would be welcome.

(2) An interesting thought is that the more rigid yeast Ess1 can be replaced by the human Pin1 but not the other way around. Is there any experimental evidence for this (in a mouse model or related)? This is just out of interest.

(3) In Figure 5C, the authors show a non-linear titration behaviour for the R30 cross peak when the 4R peptide is titrated in the solution. Do other residues show the same behaviour? And as this has been described in great length by P Wright and colleagues, can the authors translate this into a molecular picture for the WW/catalytic domain behaviour? Notably the linker region should become constrained in their model, and this should show up in the NMR spectra.

key to the transmission of the signal are not conserved. Thus, while the experiments suggested by Reviewer #1, specifically the use of NMR spectroscopy to examine *cis/trans* isomerization with different peptides, could potentially be very interesting, they would not contribute to understanding of the Pin1 allosteric model. Such experiments would also be quite challenging because the sequences of the individual peptide motifs at the two ends of our multi-repeat CTD substrates are identical, which would make it difficult to distinguish unique proton resonances for the proline residues, as we did using single-repeat peptides (Gemmill *et al.*, JBC 2005). Finally, as the reviewer pointed out, in our study we are focused on *binding* interactions (substrate targeting) rather than *cis/trans* catalytic mechanisms, which is the next logical step for ongoing studies based on data in this paper. We modified the text to re-emphasize that the mechanism in the movie is a *model* based on binding properties. The Abstract and last paragraph of Introduction now use the word "*suggest*" rather than "*indicate*" (lines 55, 135-136).

(2). This is a great idea, and we hope to do this in future studies. We did make an initial attempt in which we replaced the Ess1 linker with that of human Pin1 and to our surprise, it did not complement in yeast. However, control experiments (Westerns) showed that the chimeric protein was unstable, so this line of experiments will require additional controls and strategies. The reverse experiment would be even more interesting, *i.e.*, to show that a rigid linker would not work in higher eukaryotes like mice. This has not been done, but we would like to try it in *Drosophila* first. We thank the reviewer for their suggestion.

(3) Yes, only very few other residues showed this behavior, or other unique chemical shift trajectories unlike those observed for 1R and 5R peptides. All of these residues are shown in Fig. 5C (residues 30, 35) and in Supplementary Fig. 9 (residues 49, 130, 143) and indicated in the text (lines 361-366). We mapped these residues onto the Ess1 structure at the top of Supplementary Figure 9; we note these residues are scattered across both WW and PPIase domains, including one residue in the linker (L49). We interpreted these data to indicate subtle changes across the WW, linker, and PPIase domains to accommodate binding to the 4R peptide *vs.* the longer 5R peptide. Only one residue (L49) is in the linker domain but that is within the highly-structured α -helix. Given our crystal structure and NMR data (Figs. 1&2) showing a fully-ordered linker, and that most residues in the linker do not show non-linear behavior upon binding, we do not think the types of IDP->folded transition described by the Wright lab for c-Myb upon substrate binding (Arai *et al.*, PNAS, 2015) applies to the Ess1 linker. At present, we do not know the exact nature of the transitions given our limited data, but will be the focus of our followup work as briefly mentioned in point 1 above. We also realize that the

Reviewer #2 (Remarks to the Author):

The manuscript by Namitz and colleagues describes the structure of the Ess1 prolyl isomerase (PPIase) from *S.cerevisiae* determined by X-ray crystallography. The authors investigate the flexibility between the N-terminal WW and the C-terminal PPIase domain by small angle X-ray scattering, NMR spectroscopy, and analytical ultra-centrifugation. They analyze the binding affinity of Ess1 for the CTD hepta-repeats of the RNA polymerase II using fluorescence anisotropy, sedimentation velocity aUC, and titration NMR spectroscopy by applying various length for the substrate peptide from 1 to 5 repeats. They find that five repeats are the minimally required optimal length for binding to the two substrate recognition sites of the Ess1 proteins. Using molecular model approaches they provide a model of the flexible CTD covering the distance between the pS5P6 recognition site at the WW domain at the N-terminus and the second pS5P6 recognition site at the PPIase domain at the C-terminal end of the CTD peptide. Ess1-catalysed proline isomerization of the CTD peptide therefore requires a "lag hepta-repeat" length of 4 repeats, as this is the optimal distance between the recognition site and the active site.

This is a very nice and technically very comprehensive study to challenge the molecular characterization and structure determination of a two-domain protein containing two similar peptide recognition sites with an intrinsically unstructured peptide as the CTD, that is built of a repetitive hepta-repeat structure. The authors apply not only X-ray

cartoon in Fig. 4F might have given the false impression that the "strain" on the protein binding to 4R was primarily in the linker region. We revised this figure to indicate the sites of non-linear behavior are across the whole Ess1 protein, including the WW and PPIase domains. We also revised the text (lines 359-367) to clarify these points. We apologize for the confusion.

Summary

In addition to summary comments by Reviewer 1, this reviewer indicates the comprehensive nature of the work using multiple methodologies. Reviewer 2 has only minor comments, mostly to clarify language in the text and to improve the figures (as well as keenly identifying some critical typos). We thank the reviewer for their positive feedback and critical suggestions that have improved the clarity of the manuscript.

crystallography for the first structure determination of the ScEss1 protein, but also NMR spectroscopy with a full protein assignment and dynamic studies, small angle X-ray scattering, analytical ultracentrifugation, fluorescence anisotropy and molecular modelling.

I have only some minor comments and suggestions before recommending the manuscript for publication.

Comments:

(4) Maybe I got it wrong but the authors say in the abstract that “the Ess1 WW domain anchors the distal end of the CTD substrate during isomerization”. However, Fig. 7 B shows the N-terminus of the 5 hepta-repeat CTD peptide binding to the WW domain whereas the fifth repeat at the C-terminus is at the PPIase domain. So, the proximal part of the CTD interacts for recognition with the WW domain while four repeats further down in the distal part occurs the isomerization. Please clarify.

5) Along the line: The Movie Graphic on page 3 should indicate the N- and C-terminus of the CTD peptide. I also suggest making the background white and labelling the WW and PPIase domain to better grasp the picture. As only the backbone of the CTD is shown, maybe the backbone of the tyrosine-1 position could be labelled in a different colour to help identifying the five repeats in the recognition sequence.

(6) In the Introduction section, the authors explain the differences in the flexibility of the linker sequence between CaEss1 and human Pin1 which might impact on the engagement with substrates (lines 114-122). When explaining the new ScEss1 structure in the Results, it would be very nice to shown a superimposition of all three full length isomerases ScEss1, CaEss1 and hsPin1 that is made only on the PPIase domain, to show how differently the WW domain assembles relative to the PPIase.

(4) Indeed, we wrote "distal" which, taken literally could be confusing. We meant distal in the relative sense away from the other binding sites. However, to avoid further confusion we changed the statement in the Abstract to say "proximal" (line 55), which then jives with the figures in the text. Thank you for pointing out this confusing statement.

(5) Yes, we agree, and we have made improvements to the figure as suggested. Each of the five repeats in the CTD is now highlighted with a different color. The N and C termini have now been labeled. We have updated the movie graphic legend to reflect these changes.

(6). Indeed, superimposition of the PPIase domain shows the different juxtapositions of the WW domain in ScEss1, CaEss1 and human Pin1. This is exactly what was shown (in the supplement) as Figure S2B, which may have been overlooked because it was not cited in the text until much later in the manuscript after the ScEss1 structure was presented. We remedied this by citing Fig. S2B at this point in the text (line 122).

This could be either as a small inset in Figure 1 or in the supplement.

(7) In Figure S2A, the electron density identified for proline 6 is shown (and even a little density for the P in pS5) but the model also displays a pS2. According to the text, only one phosphorylation is present in the peptide. Please change or clarify. Also, N- and C-term of the peptide should be indicated.

(8) I don't understand how the RMSD values for the PPlase and the WW domain were determined. Both values sound very high to me when looking at the Figures 1B and 1C. First, the three chains are really difficult to identify due to the very similar blueish/greenish colors. Second, if this would be only an alignment on the Ca atom position, I suppose the RMSD values would be much smaller – to me, it looks like 0.6-0.8 Ang. Yet, if all side chains are considered in the alignment, then 1.6 and 1.7 might be reasonable. However, then also the entire molecule should be shown with all side chains.

(9) Fig. 3 would very much benefit from showing at the top the 5 peptides used in this analysis. This could be either simplistic just indicating the site of the fluorescent label and the repeats with the phosphorylation site, or in a letter code as in Table 2.

Other Comments:

(10) Abstract, line 50: As the authors speak of *S.cerevisiae* Ess1, they may refer to the 26 CTD hepta-repeats instead of writing "n".

(11) Line 99: "and 52 repeats in humans (about half consensus)". Please specify the alterations from the consensus sequence that occur mostly at position 7 from the

(7). We apologize for the confusion in Figure S2A. What we had done here is model a doubly phosphorylated single repeat CTD peptide (pSer2, PSer5) used in a co-crystal structure from a prior study with Pin1 (Verdecia *et al.*, Nat. Struct Biol 2000) in order to identify where exactly the proline 6 would reside in our structure. Indeed, there was precise overlap of the density, and our crystallographic data are consistent with NMR data on peptide binding to the WW domain. To clarify the figure, we now (1) identify the peptide as being from a Pin1 study and (2) by explaining more explicitly what was done to make the figure and provide a detailed explanation in the Fig S2A Legend.

(8) Our apologies as this was a typo left over from an earlier draft. The RMSD values were based on C-alpha and were NOT 1.6 and 1.7 Å, but rather < 0.7 Å in both cases. This has been fixed on pg. 6 (line 157) and (Cα)-designations added to the legend.

(9) Done. We agree and we clarified this in two ways. First, Fig.3 was modified to indicate the "competition" nature of the experiment, and second, we modified the Fig. 3 Legend to state exactly what peptides were used, and what positions the FITC was, and where the phosphorylation sites were (lines 672-676). This added information in the legend was necessary as it was not feasible to put of it in graphic form in the Figure itself. With these changes, we think the figure is much clearer.

(10) Done.

(11) Done, (lines 100-102). "*In humans, the divergence is most pronounced in the second half of the CTD, where substitutions at position 7 are most frequent (S>K). Despite this divergence the two S-P motifs are nearly invariant.*"

hepta-repeat in a short comment.

(12) Stoichiometry of protein-peptide (5R) binding (lines 300-307): This question might be most elegantly addressed by isothermal titration experiments. The dissociation constant (about 10 μ M for 5R) should be sufficient to get good signals. However, such analysis is beyond this study as the study is already very comprehensive with a multitude of techniques applied.

(13) But the author could more clearly discuss that not only 1 protein could bind 2 peptides (in case of the short 1R or 2R), but also 1:1 binding for 4R and 5R, while even 1 peptide could bind two proteins (5R). I personally consider the last complex formation as very unlikely and I think the data convincingly show 1:1 binding for the long peptides by aUC.

Remark:

(14) Nothing for this study, but I like to encourage the authors to try co-crystallization of the ScEss1 protein with a 5-mer CTD peptide (first and last repeat with a Ser5 phosphorylation) and possibly even a sixth repeat at the C-term, as there seems to be still some coverage. Maybe ligand soaking would already work.

(15) And did the authors test if a K7 residue changes the affinity for the recognized peptide? It would be very good to know how CTD alterations from the consensus sequence influence proline isomerization. Thanks!

(12) We agree that ITC studies could potentially be helpful, but we also agree that it would be better served in a follow-up paper.

(13). Yes, we agree with the Reviewer, and think that our SV-AUC data convincingly rules out the possibility of 1 peptide binding to two Ess1 proteins, but for the smaller peptides, the idea that 2 peptides could bind 1 protein cannot be formally ruled out. We now discuss this possibility explicitly for 1R and 2R peptides as per the reviewer's suggestion (lines 308-310).

(14). Indeed, we would like nothing more than to have a co-crystal structure with 5R CTD peptide! We have tried, but so far we have been unsuccessful in generating co-crystals. Soaking will be our next step.

(15) Testing K7-substituted peptides is an intriguing idea that we did not pursue. If that substitution alters affinity/isomerase activity of Ess1/Pin1 for the adjacent S_5 - P_6 motifs, that would be a major find!

This idea is not without precedent. For example, the Litchfield lab (Innes *et al.* Front Physiol., 2013) found evidence that the PPIase domain not only helps binding of the FL-Pin1, but also has certain sequence preferences for the position +1 to S_5 - P_6 motifs (for example, it disfavors a +1 Proline). Given that the (K7) lysine in the human CTD sequence is thought to be modified by acetylation and methylation (Eick lab, Voss *et al.*, Transcription, 2015), this modification may impart effects on binding/isomerization. We will focus on these effects as part of a followup study.

REVIEWERS' COMMENTS:

Reviewer #1 (Remarks to the Author):

The authors provide a satisfactory answer to the different questions raised. The manuscript hence can be published as is.

Reviewer #2 (Remarks to the Author):

This is a thoroughly prepared manuscript revision that addresses all points raised from the first review process. I think that particularly the movie graphic improved a lot, as well as the superimpositions, and I am happy that the RMSD value and proximal-distal discrepancies could be sorted out by the authors.